# Phosphoproteomics of osimertinib-tolerant persister cells reveals targetable kinase-substrate signatures

Hsiang-En Hsu[1,5], Matthew J Martin[2,5], Shao-Hsing Weng [1], Reta Birhanu Kitata [1], Srikar Nagelli [2], Chiung-Yun Chang[3], Sonja Hess [3✉] & Yu-Ju Chen [1,4✉]

## Abstract

Osimertinib is the first-line therapy for EGFR-mutated non-small cell lung cancer, but acquired resistance emerges in most patients and remains a major barrier for complete cure. This phenomenon is most likely associated with the drug-tolerant persister (DTP) cell phenotype, a reversible state that enables survival under treatment and leads to irreversible drug resistance. To uncover the molecular mechanism driving this distinct phenotype, we applied data-independent acquisition mass spectrometry (DIA-MS) to establish the dynamic proteomic and phosphoproteomic landscape in the osimertinib DTPs. While osimertinib initially blocks EGFR signaling, ribosome synthesis and protein translation related pathways arise in DTP phase, and resistance developed through the reactivation of EGFR downstream pathways and anti-apoptotic mechanisms such as YAP1 and mTOR-BAD hyperphosphorylation, as validated by growth combination assays. Kinase enrichment revealed elevated phosphorylation of multiple CDK1 substrates in DTP phase and pharmacological or genetic inhibition of CDK1-mediated SAMHD1 activation significantly impair DTP growth and survival. This study illuminates the dynamic landscape underlying the DTPs biology and identifies biomarker and new targets to potentially prevent or delay the onset of resistance.

**Keywords** Drug Tolerant Persister (DTP); Proteomics; Phosphoproteomics; Non-Small Cell Lung Cancer (NSCLC); EGFR
**Subject Categories** Cancer; Proteomics

## Introduction

A majority of non-small cell lung cancers (NSCLCs) harbor driver oncogenes, many of which can be targeted with clinically approved therapies. A significant breakthrough in treating NSCLC patients has been the discovery of osimertinib, a third-generation EGFR tyrosine kinase inhibitor (TKI). Osimertinib specifically targets activating mutations in the EGFR gene, including cases with or without the T790M gatekeeper mutation (Cross et al, 2014). Despite its proven clinical benefit in the metastatic setting, nearly all patients eventually relapse during drug treatment. Recent work has shown that a very heterogeneous set of genetic and non-genetic events can drive this clinical resistance (Leonetti et al, 2019), making the development of broadly applicable treatment strategies for resistant patients highly challenging. This has led to an intensified effort to characterize tumors at the minimal residual disease (MRD) phase, which is a stable period after the initial drug response where a subset of tumor cell survives but are largely dormant and cannot proliferate in the presence of the drug. MRD can be modeled in vitro via prolonged exposure of drug-sensitive parental cell lines to targeted agents, generating so-called "Drug Tolerant Persister" (DTP) cells (Sharma et al, 2010). It is now known that DTP cells represent a reversible cellular state that is characterized by dramatic changes in chromatin regulation and subsequent gene expression patterns. For example, we and others have shown a significant upregulation of RNAs controlled by the YAP-TEAD complex in DTPs, and accordingly DTP cells are sensitive to small-molecule TEAD inhibitors (Criscione et al, 2022; Kurppa et al, 2020), exemplifying a novel vulnerability that could potentially be exploited in the clinic to delay the emergence of resistance. To date, less attention has been paid to the functional output of these transcriptional changes, namely the unique landscape of proteomic and phosphoproteomic alterations that accompany the drug-tolerant state.

The emergence of mutations and molecular adaptations that alter functional pathways and induce histological changes can drive the development of drug resistance. In particular, aberrant protein kinase activity and the subsequent changes in protein phosphorylation have been strongly linked to tumorigenesis and treatment resistance. Proteomic and phosphoproteomic analysis offer an unprecedented opportunity to uncover the dynamic, systemwide changes of the signaling networks that contribute to resistance, extending beyond the scope of genomic approaches. Such comprehensive (phospho)proteomic analysis enables the elucidation of functional heterogeneity and the mechanism that determines how therapeutic resistance can be overcome. Therefore, these characteristics have attracted attention as potential therapeutic targets, aiming at treating cancer and reducing treatment resistance (Bhullar et al, 2018; Franciosa et al, 2021; Giamas et al, 2007).

[1]Institute of Chemistry, Academia Sinica, Taipei, Taiwan. [2]Oncology Targeted Discovery, Oncology R&D, AstraZeneca, Cambridge, UK. [3]Dynamic Omics, Centre for Genomics Research, Discovery Sciences, BioPharmaceuticals R&D, AstraZeneca, Gaithersburg, MD, USA. [4]Department of Chemistry, National Taiwan University, Taipei, Taiwan. [5]These authors contributed equally: Hsiang-En Hsu, Matthew J Martin. ✉E-mail: Sonja.Hess@astrazeneca.com; yujuchen@as.edu.tw

To investigate the altered signaling pathway and regulatory network related to DTP status, we employed a library-enhanced data-independent acquisition mass spectrometry (DIA-MS) to conduct complementary proteomic and phosphoproteomic analyses on an osimertinib DTP model, which allows us to map the dynamic landscapes during the transition from acute drug responsiveness to drug tolerance. Towards deep profiling of the DTP network and key signaling pathways, we established a DTP-specific proteome spectral library (12,360 protein groups and 256,941 peptide sequences) along with a phosphoproteome spectral library (221,618 phosphopeptides corresponding to 53,182 phosphosites on 10,326 protein groups). The dynamic profiles revealed previously under-explored dynamic signaling networks that characterize the molecular transition from acute phase to DTP state, potentially underpinning resistance mechanisms. Further validation experiments verified biomarker changes in DTPs, which were then leveraged to target key DTP signaling pathways. Notably, we show here that targeting CDK1 genetically or pharmacologically can impair DTP growth and survival. These findings identify CDK1 as a crucial signaling node in the emergence of drug resistance in NSCLC following osimertinib treatment. Furthermore, the DTP spectral library is made publicly accessible to provide a comprehensive resource for advancing proteomics and phosphoproteomics applications in cancer research.

## Results

### Osimertinib DTP model and experimental design

In this study, a DTP model was established by using a non-small cell lung cancer cell line (PC9; Exon19 deletion) (Criscione et al, 2022). For our analysis, we explored two cellular states, (1) treatment with osimertinib for 24 h to mimic initial drug response (acute phase); and (2) prolonged exposure for 21 days to generate DTP cell subpopulation (DTP phase). To study the recovery of DTP cells after drug withdrawal, DTP cells were washed and incubated with drug-free media for either a short-term (24 h, short washout) or long-term (7 days, long washout) duration (Fig. 1A), which we have previously shown results in a partial re-sensitization to drug treatment (Martin et al, 2022).

To achieve quantitative proteomic and phosphoproteomic profiling of the different cellular states, we performed DIA-MS to investigate these dynamic regulations. We previously demonstrated that high-quality reference spectral libraries greatly enhanced the identification, coverage, and sensitivity of a DIA-based label-free approach (Kitata et al, 2021). For this study, we constructed two customized hybrid spectral libraries for the proteome and phosphoproteome. These libraries were developed by incorporating an additional 36 DIA datasets from proteome and phosphoproteome analyses into our original spectral libraries, which were constructed from NSCLC cancer cell lines and patient tissues (Fig. 1B). Specifically, these additional datasets were obtained by DIA-MS analysis from triplicate runs for each time point at acute and DTP phases. To generate the hybrid phosphopeptide reference library, the database search results from data-dependent acquisition (DDA) and DIA datasets were performed by Spectronaut Pulsar, to ensure identification confidence of a 1% false discovery rate (FDR) at PSM, peptide, and protein level as well as for the estimation of

phosphosite localization probability (≥75%). In total, the proteome and phosphoproteome spectral libraries cover 352,022 unique peptide precursors from 12,360 protein groups and 221,618 phosphopeptides from 10,326 proteins, respectively. Lastly, single-shot MS analysis was performed for individual samples from different time points using in-house hybrid spectral libraries for identification and quantitative comparison of the protein expression and site-specific phosphorylation (Fig. 1C).

### Quantitative (phospho)proteomic profiles reveal distinct acute and tolerant phases

To explore the dynamic alteration of protein expression and phosphorylation events in response to osimertinib in both the acute and DTP phases, two sets of proteomic and phosphoproteomic experiments were performed by DIA-MS for the following conditions (1) DMSO (vehicle control), (2) Acute phase (osimertinib treatment for 5 min, 10 min, and 6 h), and (3) DTP phase (DTP, DTP recovery for 24 h and 7 days), followed by library-based searching and data deconvolution. Phosphopeptides were further enriched by iron-based immobilized metal affinity chromatography (Fe-IMAC). Lung cancer proteome and phosphoproteome mass spectra libraries were established to enhance profiling coverage by integrating all DDA raw files searched by MaxQuant with an FDR of 1% at PSM, and protein group. The results reveal that similar numbers of proteins (an average of 5200) and phosphopeptides (an average of 21,500), were identified from each time point treatment (Appendix Fig. S1A,B). A total of 2953 proteins exhibited significant differential expression in the proteomic analysis. For the phosphoproteome, a total of 10,742 unique phosphosites from 38,957 phosphopeptides were identified from 3625 proteins. For quantitative comparison, batch correction and normalization were performed by subtraction mean of total abundance across all proteome and phosphoproteome dataset batches (Appendix Fig. S1C). Between these two datasets, 1006 proteins overlapped, indicating that 34% of the total identified proteins could be quantified in terms of phosphorylation and protein levels (Dataset EV1).

To evaluate the impact of osimertinib treatment, we initially performed a principal component analysis (PCA) using batch-corrected data to compare the proteome profiles across different time point treatments. Both proteome and phosphoproteomic profiles show that three phases (DMSO control, Acute, and DTP) are clearly separated (Fig. 2A, proteome part; phosphoproteome part shown in Appendix Fig. S2). The biological replicates of each treatment condition cluster together, indicating good reproducibility. Notably, after the long-term recovery post drug washout, the DTP conditions have relatively limited scattering distributions and are clearly separated from both DMSO and the acute phase on Principal Component 1 (PC1). PC1 represents the most variance (52.3%) powered by the DTP samples, while Principal Component 2 (PC2) contains the second most variance (13.4%) driven by the samples upon osimertinib acute treatment.

To gain insights into the biological characteristics of the DTP state in PC9 cells, we performed an ANOVA test (Benjamini–Hochberg, FDR < 0.05) and identified 2953 and 3711 differentially expressed proteins (DEPs) and phosphopeptides among the three conditions, respectively. Hierarchical clustering of these DEPs reveals the similarity and significant up- and downregulation patterns at different cellular states (Fig. 2B). Interestingly, the proteome and phosphoproteome heatmap shows similar kinetic expression profiles with two

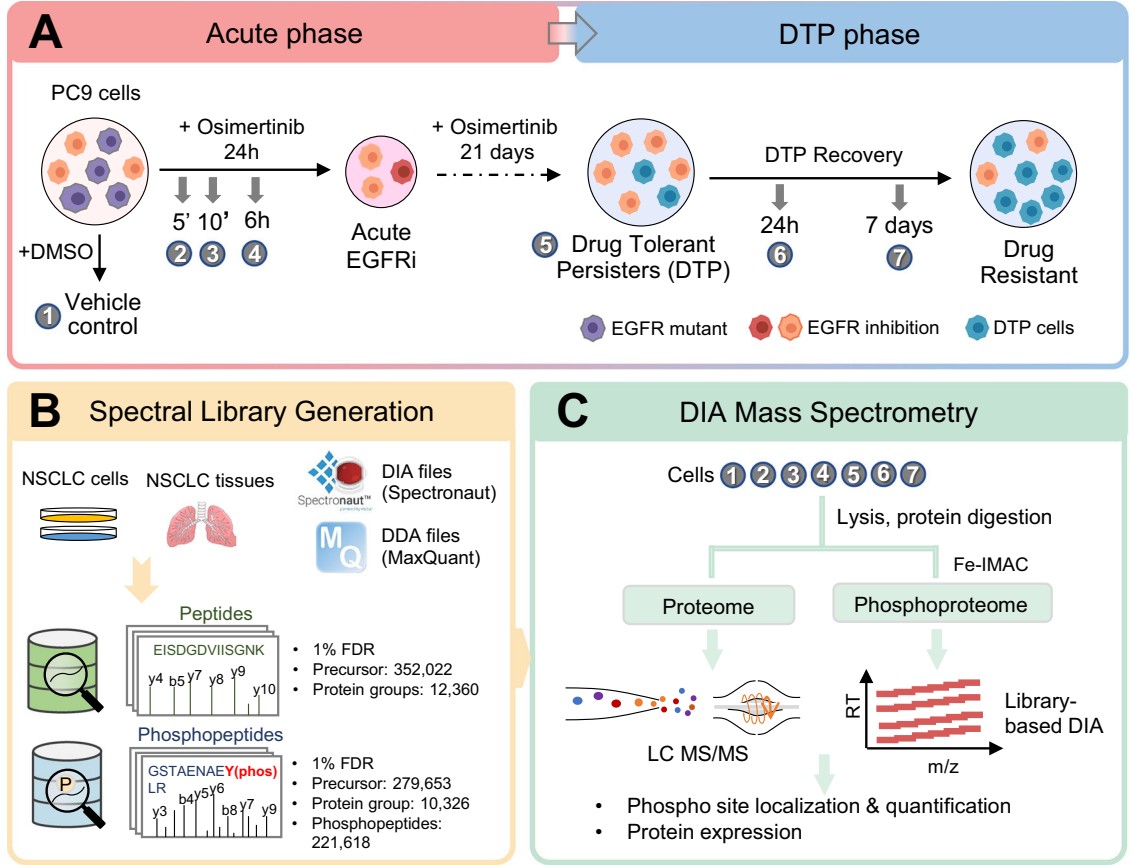

**Figure 1. Phosphoproteomics DIA analysis workflow.**

(A) PC9 cells were treated ($n = 3$) with osimertinib (160 nM) acutely (5 min, 10 min, and 6 h) or for 21 days to generate DTPs. In addition, the drug was washed out of the cells for the recovery phase (24 h, and 7 days). After treatment, proteins were extracted from the cell lysates and were digested with Trypsin/Lys-C. (B) Customized proteome and phosphoproteome spectral libraries for lung cancer were constructed from both DDA and DIA datasets, which were processed by MaxQuant and Spectronaut Pulsar, respectively. (C) For the phosphoproteome, digested lysates (~200 µg) were enriched for phosphopeptides by Fe-IMAC approach prior to LC/MS-MS analysis. Proteome and phosphoproteome data were acquired in DIA mode and analyzed against in-house constructed hybrid spectral libraries.

clusters of opposite expression trends. The upper cluster reveals the relatively similar pattern between control (DMSO) cells and the acute phase cells with a small subset of downregulation in the acute phase, which likely represents the osimertinib-responsive inhibition targets. The lower cluster shows the upregulated pattern in the DTP state, with or without drug-free recovery, which may represent adaptive factors allowing for survival under prolonged osimertinib treatment. To decipher the biological processes of these responsive events, biological process enrichment was performed against the KEGG database. In addition to the significant enrichment in metabolic pathways at the proteome level (Fig. 2C), both results indicate that actin cytoskeleton, cell adhesion, tight junction, and endocytosis pathways are significantly and commonly upregulated in DTP cells ($P < 0.05$) (Fig. 2C,D), suggesting potential membrane remodeling in the DTP state.

## Unraveling the proteomic and phosphoproteomic landscape of osimertinib-driven drug resistance and recovery

Next, we sought to explore the signaling pathways associated with DTP cells. Comparing the DMSO control with the acute phase

(represented by the ratio of acute/DMSO), it is expected that osimertinib-responsive targets would be downregulated during the acute phase. We further hypothesized that targets enabling osimertinib tolerance might show either elevated protein expression or overly activated phosphorylation status in the DTP phase. Thus, protein expression ratios differentially regulated under three DTP conditions (DTP, DTP-24h, and DTP-7d) compared to the average of acute phases, may reflect constitutively activated or repressed targets driven by prolonged drug treatment. A quantitative proteomic comparison identified 3396 DEPs, which were further classified into 6 clusters with different alteration trends by hierarchical clustering followed by annotation of biological processes using the KEGG database (Kanehisa et al, 2016) (Fig. 3A,B). Two clusters showing upregulation with long-term DTP recovery (7 days) may represent constitutively activated events, such as DNA repair and RNA splicing enriched in cluster I and cluster II (continuously up in DTP-7d) associated with drug metabolism. The result is consistent with a previous report that DNA repair is compromised in osimertinib-resistant cells and inhibition of DNA-PK, a key kinase mediating NHEJ repair, enhances DNA damage and recovers the sensitivity of the resistant

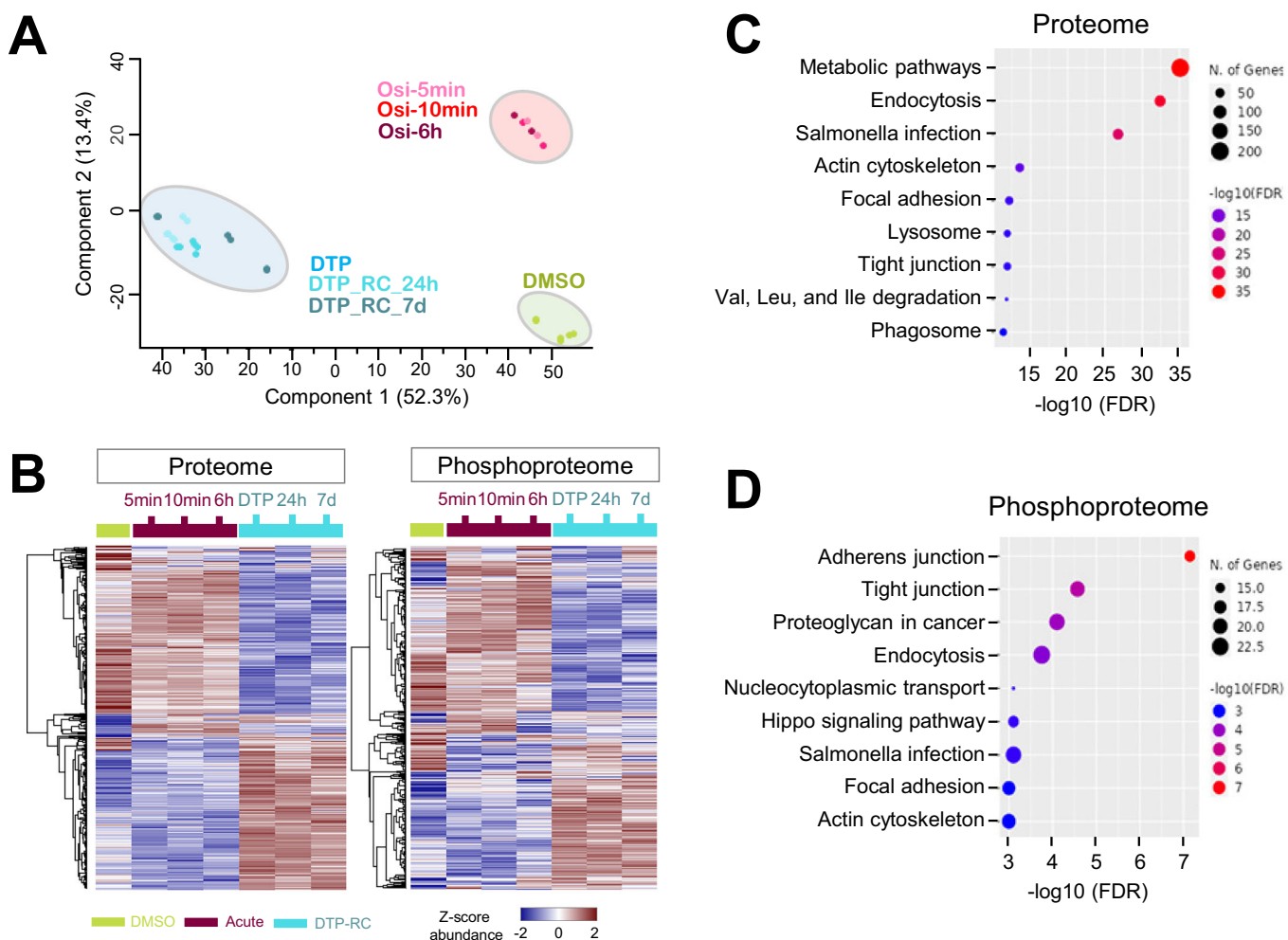

**Figure 2. Proteome and phosphoproteome profiling in response to drug treatment and recovery.**

(A) PCA plot for proteome data. (B) Hierarchical clustering of differentially expressed proteins between the proteome and the phosphoproteome. (C) KEGG pathway analysis of differentially expressed proteins (D). KEGG pathway analysis of differentially expressed phosphopeptides.

cells to osimertinib (Liang et al, 2021). Intriguingly, two clusters (clusters V and VI) exhibited significant enhancement in short-term recovery before reverting to their baseline levels during long-term recovery. These clusters are highly correlated with protein regeneration (ribosome biogenesis, cytoskeleton, and translation), suggesting for cellular recovery that allows DTP cells to resume proliferation once the drug pressure is relieved, contributing to long-term drug resistance and potentially driving cancer pathogenesis (Pelletier et al, 2018). Volcano plot analysis comparing the ratios of acute/DMSO and DTP-7d/acute conditions reveal tens of significantly downregulated proteins in the acute phase compared to DMSO, likely presenting putative drug-inhibited targets (Fig. 3C), as well as substantially upregulated proteins in the DTP phase which are likely the drug-resistant targets induced by long-term osimertinib exposure (Fig. 3D). Among them, we note that many of these proteins, SERPINF2, AURKB, A2M, ALB, PZP, ITIH2, are related to ribosome synthesis and protein translation (shown in yellow) with a trend of inhibition by osimertinib treatment while elevation in the DTP phase. Though these are not

classical ribosomal proteins, the involvement of SERPINF2, AURKB, A2M, and other proteins in ribosomal biogenesis pathways not only suggest for cellular adaption to stress and recovery, the association of AURKB with the key regulators of mTOR, may contribute to the activation of survival pathways that help DTP cells maintain minimal metabolic activity and evade drug-induced apoptosis (Tanaka et al, 2021). Consistent with the previous report, significant upregulation of AURKB (Fig. 3D), is associated the DTP state (Criscione et al, 2022).

To understand critical signaling events that accompanied chronic osimertinib treatment, we monitored the dynamic protein phosphorylation state of DTP cells compared to both the DMSO control and the acute phase. Within the NSCLC-related signaling pathways, supervised hierarchical clustering analysis for the differential phosphosites across all time points showed differential regulation in MAPK, PI3K/AKT, PKA, and PKC signaling (Fig. 4A). Though osimertinib effectively blocked the EGFR signaling pathways and several downstream nodes during the acute phase, as expected, the inhibition also led to dynamic

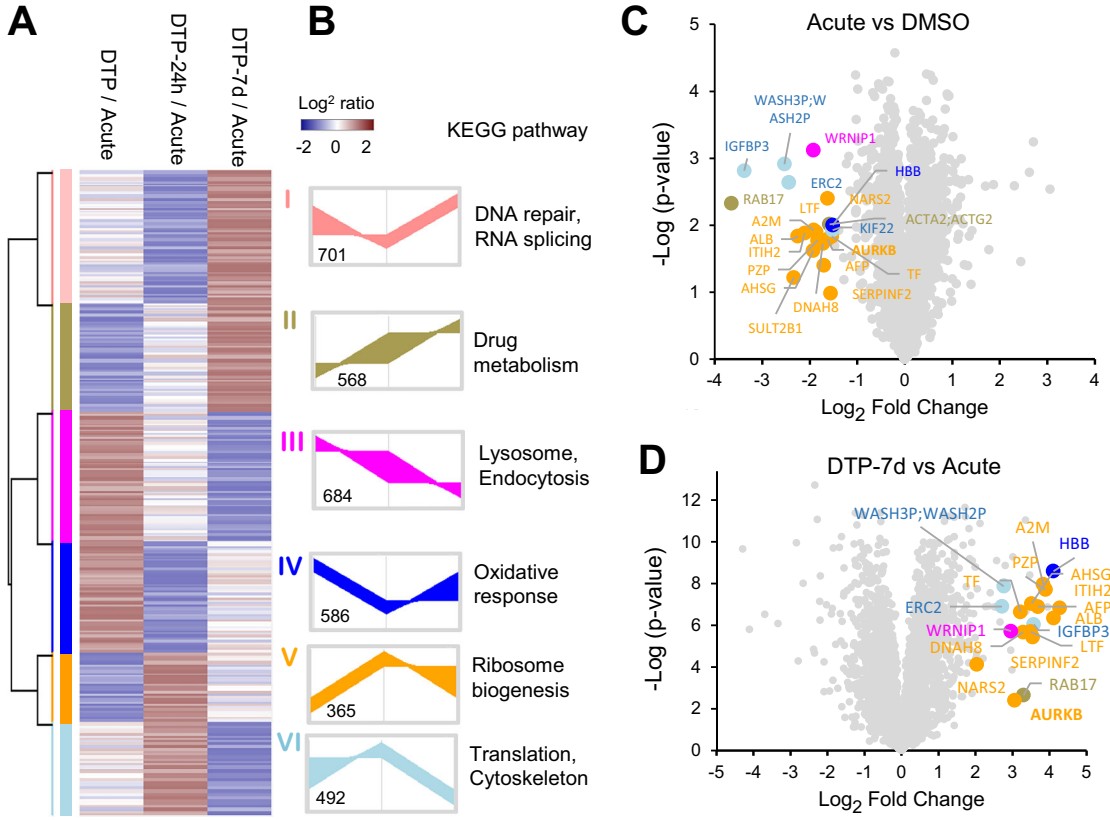

**Figure 3. Protein abundance comparison in the DMSO, acute and DTP recovery phases.**

(A) Hierarchical clustering of proteins differentially expressed between acute, DTP, and DTP recovery phases. (B) KEGG pathway enrichment analysis of protein clusters from (A), highlighting representative biological processes. (C) Volcano plot comparing acute vs. DMSO-treated control, showing significantly altered proteins (t test, Benjamini–Hochberg FDR < 0.05, $n = 3$ biological replicates per group). Twenty candidate proteins were significantly downregulated in the acute phase. (D) Volcano plot comparing DTP-7d recovery vs. acute phase, showing upregulated proteins during recovery (t test, Benjamini–Hochberg FDR < 0.05, $n = 3$ biological replicates per group).

changes in phosphorylation within key oncogenic signaling pathways. During the acute phase, phosphorylation levels of most key components were reduced, indicating pathway suppression. However, as cells transitioned into the DTP and DTP recovery phases, phosphorylation gradually increased (Fig. 4A), suggesting reactivation of these pathways. The inhibition of EGFR activity significantly suppressed downstream phosphorylation cascades such as PI3K/AKT (GAB1 at S266, S355, Y373, and T503; mTOR at S2478 and S2481) (Fig. 4B; Appendix Fig. S3B), PKA (PRKACA at T198), PKC (PRKCD at T507 and S645; PRKCG at T655; PRKCI at S247, T412, and Y419; PRKCQ at Y545; PRKCZ at Y417) (Fig. 4C), and MAPK (BRAF at T373; ARAF at S157 and S257; RPS6KA3 at Y226, Y234, and S369) (Fig. 4D; Appendix Fig. S3A). Notably, BRAF-S365 is an inhibitory phosphorylation site known to suppress MAPK pathway activation. Although most phosphorylation events lead to pathway activation, having both activating and inhibitory modifications shows that the regulation process is complicated. Among these complex phosphorylation events, some, such as GAB1-Y373, mTOR_S2481, and PKA_T198, are potentially associated with the activation of their respective pathways (PI3K/AKT, MAPK, and PKA). Thus, reduced phosphorylation of these phosphosites suggests their potential utility as

a biomarker to monitor the efficacy of osimertinib therapy. In contrast, phosphorylation of BRAF at S365 has been shown to inhibit its activity (Guan, 2000). These distinct phosphorylation events, leading to either activation or inhibition, highlight the dynamic regulation of NSCLC signaling in response to drug treatment. Interestingly, Bcl-2 agonist of cell death (BAD), a significant regulator in anti-apoptotic processes, exhibited increased phosphorylation at Ser99 and Ser118, which coincided with the upregulation of its upstream kinases PKA and PKC during the DTP phase (see heatmap in Fig. 4A,C). While BAD phosphorylation is generally associated with inhibition of its pro-apoptotic function (Datta et al, 1997), the increased phosphorylation of PKA and BAD align with our previous report that GPER-mediated PKA activation leads to BAD Ser118 phosphorylation. Thus, these events may contribute to the activation of anti-apoptotic processes, in line with the observed survival advantage of these cells (Qian et al, 2022). In summary, we show a comprehensive dynamic phosphorylation network in NSCLC signaling that reveal the re-establishment of pathways downstream of EGFR and the induced hyperphosphorylation of BAD may be involved in the mechanism to prevent cell death during DTP recovery, thereby promoting cell survival (Mann et al, 2019).

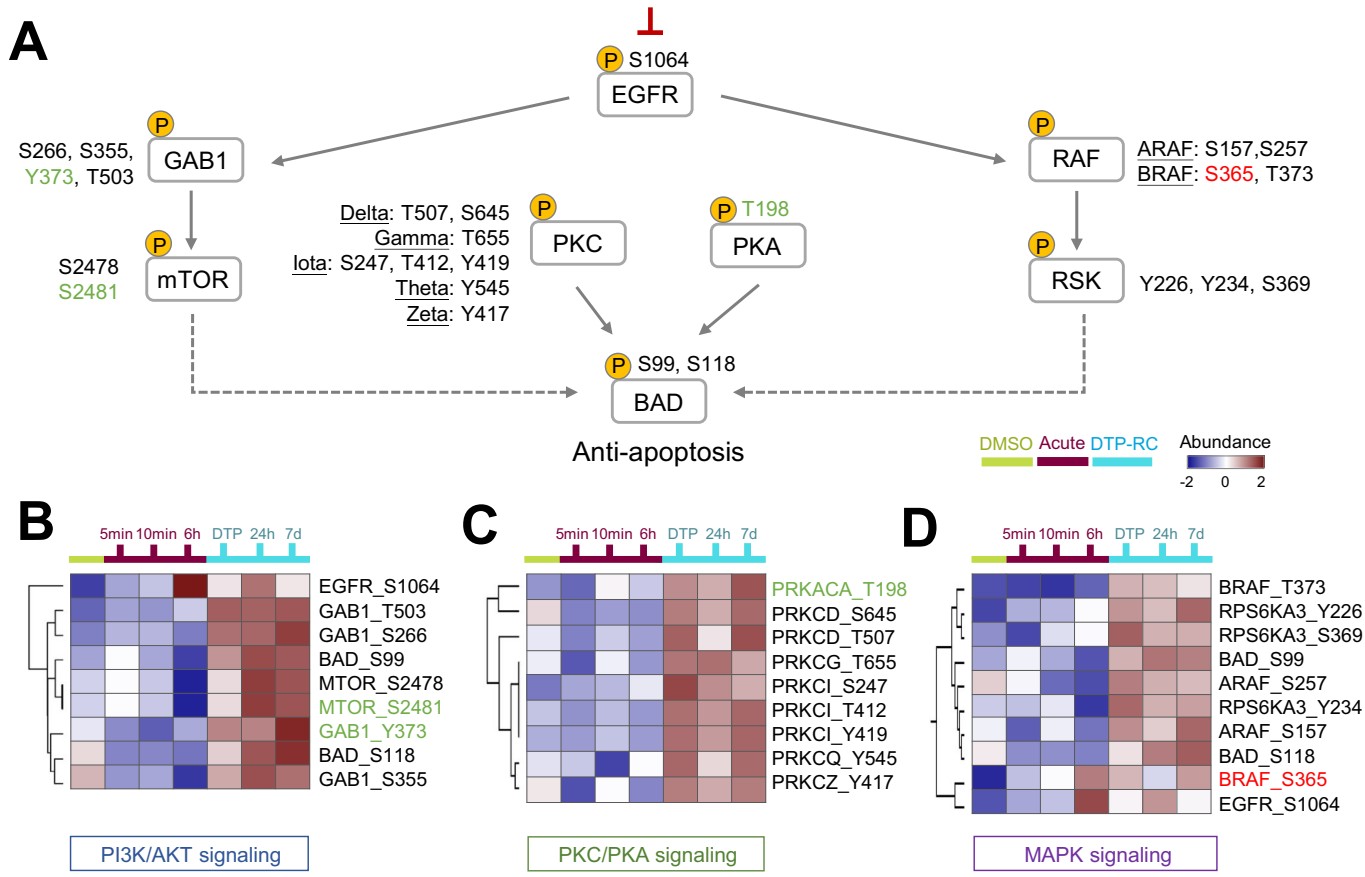

**Figure 4. Phosphoproteomic analysis of osimertinib time-point treatments in NSCLC signaling pathways.**

(A) Comprehensive dynamic phosphorylation network of DTPs in the NSCLC signaling pathway. Green labels (e.g., GAB1-Y373, mTOR-S2481, and PKA-T198) denote activating phosphorylation events, whereas red labels (e.g., BRAF-S365) indicate inhibitory phosphorylation. (B) The phosphorylation site dynamics during DTP recovery in the PI3K/AKT signaling pathway. (C) The PKA/PKC signaling pathway shows dynamics of phosphorylation changes during DTP recovery. (D) Phosphorylation dynamics during DTP recovery in the MAPK signaling pathway.

## Inhibiting CDK1 activity impairs DTP growth and survival

To study other aberrant tyrosine kinases that may be intimately linked with the mechanisms of resistance, the differential phosphosites during the DTP phase were used to extract individual substrate motifs and identify their corresponding kinases (Fig. 5A,B; Appendix Fig. S4) (Hornbeck et al, 2015). By the kinase–substrate enrichment analysis (KSEA), we showed that CDK1 activity changes during the treatment phases, with an increase in the phosphorylation of several CDK1 substrates during the DTP phase (Fig. 5A). The kinase enrichment analysis reveals that the CMGC kinase family is the largest category responsible for regulating 17 overrepresented substrates, including the prominent kinases CDK1, CK2A1, along with additional highly rated substrates CDK2, CDK7, ERK1, and GSK3B. Consistent with our findings in Fig. 4, the analysis also revealed substrate motifs and corresponding kinases associated with the PI3K/AKT pathway (GAB1-pY373, mTOR-pS2478, mTOR-pS2481, BAD-pS99, and BAD-pS118) that exhibited dramatically increased phosphorylation levels during the DTP phase (Fig. 5B). Notably, five distinct CDK1 substrates show enhanced phosphorylation in DTP, strongly indicating elevated activity of CDK1 in the DTP phase.

We next wished to confirm the finding of enhanced CDK1 activity in DTP cells compared to acutely treated parental cells using phospho-specific antibodies by western blot. Upon treatment of PC9 cells with osimertinib, we observed a slight decline in phospho-CDK1 at the tyrosine-15 inhibitory site (Y15) after 6 h and complete inhibition at the DTP stage of treatment (Fig. 6A). Interestingly, this was partially reversed by drug washout at 24 h and fully reversed with significantly increased level after 7 days without drug. The early inhibition and recovery in DTPs of CDK1-Y15 phosphorylation in DTP cells was also confirmed in the H1975 and HCC4006 cell lines (Appendix Fig. S5A,B). Importantly, we saw significant upregulated phosphorylation of CDK1 substrates SAMHD1 and PML in DTPs from the PC9 (Fig. 6B), H1975, and HCC4006 cell lines (Appendix Fig. S5C). We note that there was a corresponding increase in total protein levels of SAMHD1 and PML in DTP cells that correlated with enhanced CDK1 activity.

To determine the functional relevance of these observed phosphorylation events that suggested upregulation of CDK1 activity in DTP cells, we tested an inhibitor of CDC25-NSC-663428 on the establishment or regrowth of DTPs. CDC25 dephosphorylates CDK1 at Y15 as well as the additional inhibitory site Thr14, and thus inhibition of CDC25 serves to also inhibit CDK1. Notably, we see that while 24 h

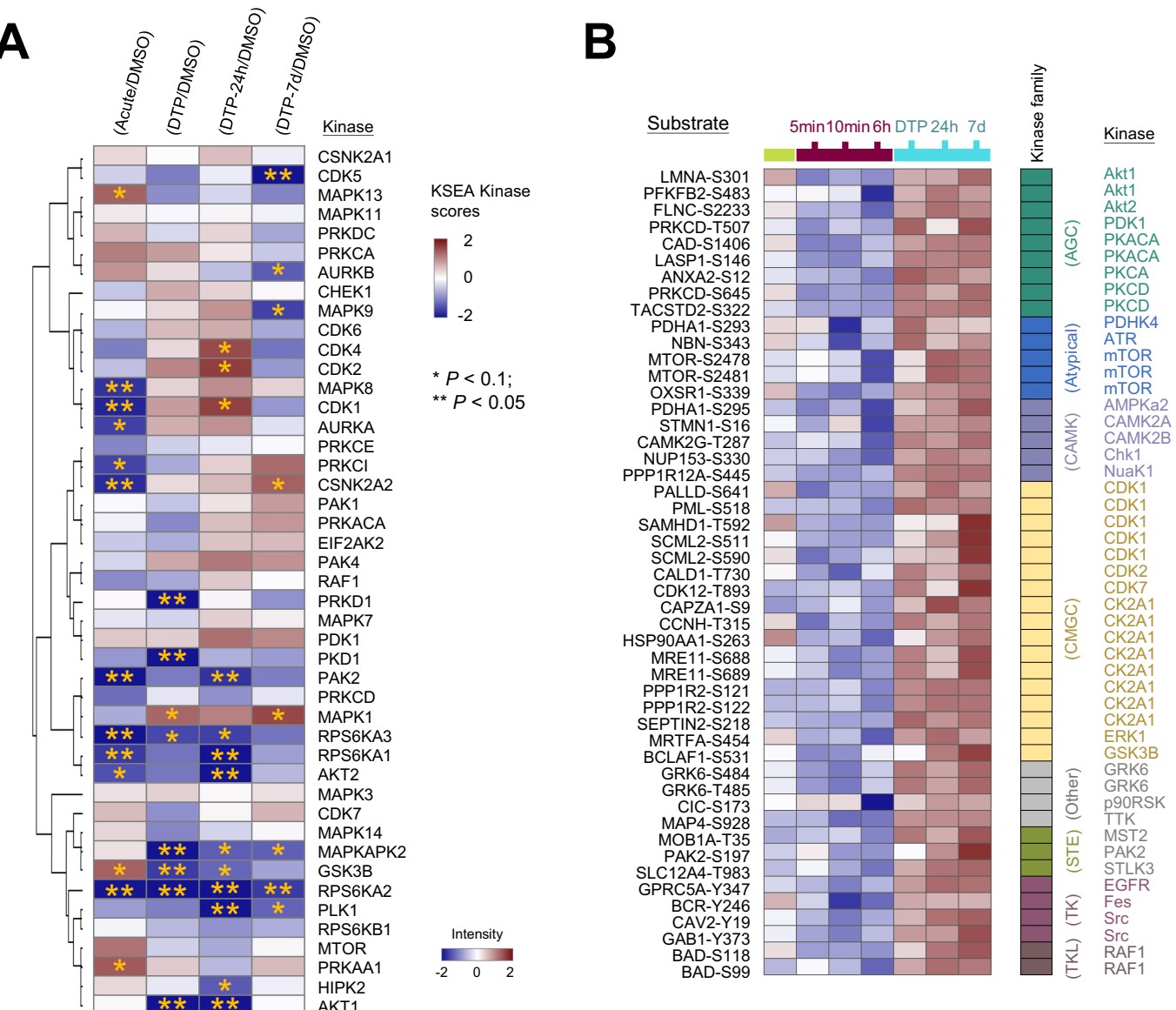

**Figure 5. Kinase enrichment analysis reveals a potential CDK regulation pathway.**

(A) Heatmap showing the enrichment of substrate groups for the different kinases calculated by the KSEA (https://casecpb.shinyapps.io/ksea/). Only kinases that are shared between the four datasets and that have 5+ substrates are included. Blue color represents negative kinase scores, and red represents positive. Asterisks indicate the scores of P values (*P < 0.1; **P < 0.05). (B) Significantly enriched phosphosites and their corresponding kinases among different time points.

osimertinib causes a reduction in Y15 phosphorylation, this can be at least partially reversed by co-treatment with the CDC25 phosphatase inhibitor NSC-663428 (Appendix Fig. S6A). We conducted experiments to access the cell growth using three distinct dosing schedules (Fig. 6C): (1) an upfront combination therapy followed by release into drug-free media; (2) generation of DTPs by a period of osimertinib monotherapy treatment, followed by a switch to the CDC25 inhibitor as monotherapy; and (3) generation of DTPs followed by a switch to the CDC25 inhibitor/osimertinib combination therapy, and finally release into drug-free media. Critically, when we treated PC9 (Fig. 6D), or H1975 (Appendix Fig. S6B) with each of the three treatment schedules, we observed significant inhibition of cell growth compared

to osimertinib monotherapy. Notably, in HCC4006 cells, a cell line known to rapidly develop osimertinib resistance in culture, co-treatment with NSC-663428 significantly delayed the onset of the osimertinib resistance phenotype (Appendix Fig. S6C).

CDK2 is among the 17 enriched kinases with elevated phosphorylation of its substrate CALD1-T730. Given the overlapping and potentially compensatory roles of CDK1 and CDK2 in cell cycle regulation (Ding et al, 2020), we sought to assess the contribution of both kinases to DTP biology. We employed CRISPR/Cas9 to knock out CDK1 and CDK2 expression in H1975, HCC4006, and PC9 cell lines (Appendix Fig. S7A). Overall, CDK1 and CDK2 knockout led to a further reduction in cell confluency compared to the osimertinib-

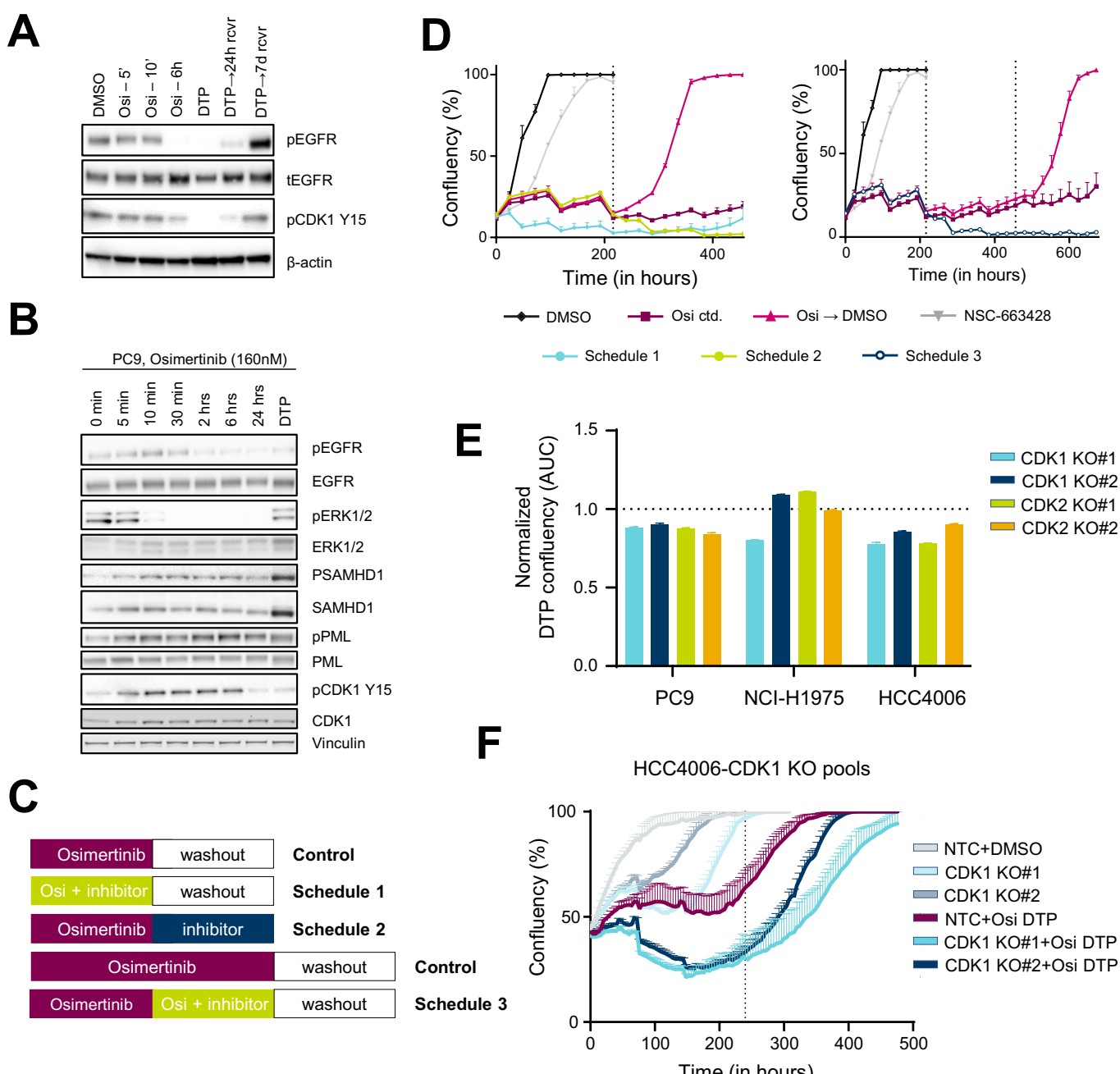

**Figure 6. The inhibition of CDK1 function with osimertinib delayed tumor regrowth in vitro.**

(A, B) Western blotting against the indicated proteins in the PC9 cell line treated with osimertinib over a range of times. (C) Schematic of DTP growth assay dosing schedules. (D) PC9 cells were treated long-term with the indicated schedules of osimertinib (160 nM) and/or NSC-663284 (1 µM). Growth was assessed as % confluence as measured by the Incucyte imaging platform. The dotted line represents the time point where dosing was changed. Error bars represent standard deviation (SD; $n = 3$). (E) Relative DTP confluence of CDK1 and CDK2 knockout pools in PC9, H1975, and HCC4006 cells from its DTP growth assays. Cells were treated with osimertinib (160 nM) for 10 days (DTP), followed by drug washout to evaluate regrowth. Growth was assessed as % confluency measured by the Incucyte imaging platform. Bar graphs represent normalized AUC of the plotted growth curves (normalized to the effect observed in non-targeted control (NTC) samples – indicated by dotted line; $n = 3$). (F) Cell confluence plot for HCC4006 cells transfected with CRISPR constructs targeting CDK1 and treated with 160 nM osimertinib for 10 days, followed by drug washout. NTC indicates non-targeted control (NTC) gRNA transfected samples. Error bars represent SD ($n = 3$). Source data are available online for this figure.

induced DTP states in these cells (Fig. 6E). In addition, when assessing the growth curves, we observed a noticeable delay in the regrowth of osimertinib-induced DTP in CDK1 and CDK2 knockout cells (Fig. 6F; Appendix Fig. S7B–D). In summary, our findings demonstrate that CDK1 activity is elevated in DTP cells, resulting in increased activation in several of its substrates, and inhibiting CDK1 function pharmacologically or genetically can impair DTP growth and survival.

## Potential regulatory signaling networks in osimertinib DTP cells

To gain a deeper understanding of the complete cellular signaling dynamics in osimertinib-induced DTPs derived from PC9 cells, we performed a network enrichment analysis using CausalPath (Babur et al, 2021). This analysis revealed the potential interplay of multiple major kinase nodes and their activating and/or inhibited phosphorylation-mediated networks (Fig. 7A). CDK1 exerts a broad impact in regulating many downstream events, such as SAMHD1-pT592 phosphorylation to mediate cell cycle regulation and PML-pS518 phosphorylation to regulate its protein stability during hypoxia-induced stress environment (Hsu and Kao, 2018). Within this network, MAPK14 has been reported to directly phosphorylate EGFR-pT693 during endocytosis (Smith et al, 2021) and is being associated with multiple events, such as regulating the phosphorylation of CDC25B-pS375, JUN-pS63, ATF2-pT71, STAT3-pS727, TP53-pS315 (Fig. 7A). Further extraction of the known kinase–substrate network centering MAPK14 show its downstream regulation on CDC25B-pS375 and EGFR-pT693, followed by coordinate control of CKD1 activity via pT14 or pY15 phosphorylation (Zhong et al, 2017) (Fig. 7B). CDK1 can positively regulate SAMHD1-T592 phosphorylation and other substrates to fine-tune cell cycle regulation (Fig. 7A, shown in green box and arrow). In addition, the result also showed an independent network of YAP1-127 phosphorylation of the Hippo pathway components, which may be activated by phosphorylation of MOB1 at Thr35 (Zeng and Hong, 2008; Zhao et al, 2007) or downregulated by dephosphorylation of AMOTL2-pS759, causing the promotion of cell proliferation and tumorigenesis (Artinian et al, 2015) (Fig. 7C, also shown in blue box of Fig. 7A).

Next, we wished to validate and characterize key signaling pathways that are highlighted as critical to DTP biology in the phosphoproteomics dataset. One notable example was the activation of mTOR, which promotes cell survival by phosphorylating BAD, thereby inhibiting apoptosis. Our analysis showed that phosphorylation of S6, an mTOR downstream target, was significantly elevated in DTP cells compared to cells treated acutely with osimertinib in both PC9 and HCC4006 cells (Fig. 7D; Appendix Fig. S8A). This phosphorylation event was attenuated significantly by treatment with the mTORC1/2 inhibitor vistusertib. The significance of this pathway in DTP biology was further validated by long-term growth combination assays, where co-treatment with vistusertib could impair DTP growth, particularly inhibiting the regrowth of established DTP (schedule 3, Fig. 7E), in three EGFR-mutant cell lines. Moreover, continuous co-treatment with vistusertib significantly delayed the rapid-onset resistance phenotype of the HCC4006 cell line (Appendix Fig. S8B). In addition, elevated YAP phosphorylation at serine 109 and 127 was observed in DTP cells. These sites are known to drive YAP binding to 14-3-3 proteins and sequestering this transcriptional coactivator

in the cytoplasm to attenuate its ability to interact with TEAD and drive gene expression (Gessler et al, 2024). Interestingly, other studies have shown a coordinated upregulation of YAP-TEAD transcriptional activity in tolerant cells (Kurppa et al, 2020; Criscione et al, 2022). To resolve this apparent contradiction, we explored YAP phosphorylation and YAP-TEAD target gene expression by western blot. The results confirmed an increase in YAP serine 127 phosphorylation as early as 48 h after osimertinib treatment. Nevertheless, in these same samples, upregulation of CYR61, a canonical YAP-TEAD target gene, was maintained at 72 h of osimertinib treatment and persisted in the drug-tolerant state (Fig. 7F; Appendix Fig. S9A,B). Moreover, we also observe increased Ser127 phosphorylation in DTPs generated from NF2 knockout cells (Appendix Fig. S9C), which have been shown to have enhanced activity in YAP1-driven transcription (Pfeifer et al, 2024), indicating that this phosphorylation event does not completely block transcriptional activity. Together, our findings indicate that mTOR and YAP1 signaling may be active in osimertinib-treated cells, but further investigation is needed to fully evaluate their role in the survival and regrowth of persister cells. The persistent upregulation of YAP-TEAD target genes in DTPs suggests a complex regulatory mechanism that allows YAP signaling to remain active in the drug-tolerant state, providing potential therapeutic targets for overcoming resistance.

## Discussion

This study offers a systems-level analysis of the dynamic proteomic and phosphoproteomic landscape using the DTP model, providing deep insights into the molecular shifts from the acute response to adaptation to sustained osimertinib treatment. Using advanced spectral library-assisted DIA-MS toward comprehensive mapping of the DTP model, this study constructed the first proteomic and phosphoproteomic mass spectra libraries specific to DTP. These libraries achieved unprecedented coverage of 12,360 protein groups and 221,618 phosphopeptides from 10,326 proteins, significantly enhancing the depth of protein expression and site-specific phosphorylation analysis. By integrating the temporal profiles from the proteomic and phosphoproteomic data, we uncovered a previously under-explored dynamic network of DTP and key signaling pathways underpinning resistance mechanisms and identified several potentially novel druggable targets, expanding insight into the role of these targets in osimertinib resistance and DTP biology, which may inspire potential areas for further therapeutic exploration.

With advantages of high sensitivity and reproducibility (Kitata et al, 2021), this study demonstrated that the DIA-MS approach enables the detection of a broader spectrum of proteins and phosphorylation modifications compared to traditional methods. This capability is particularly important, as the detection and quantification of low-abundance phosphorylation sites are often challenging but crucial for advancing biomarker discovery and offering deeper insights into disease mechanisms and treatment responses. Future biomarker development will benefit from systematic investigation of post-translational modifications such as phosphorylation, highlighting the vital role of robust proteomic platforms in translational research. With the high proteome coverage compared to the recently updated human proteome of 18,397 of the 19,778 neXtProt predicted proteins coded in

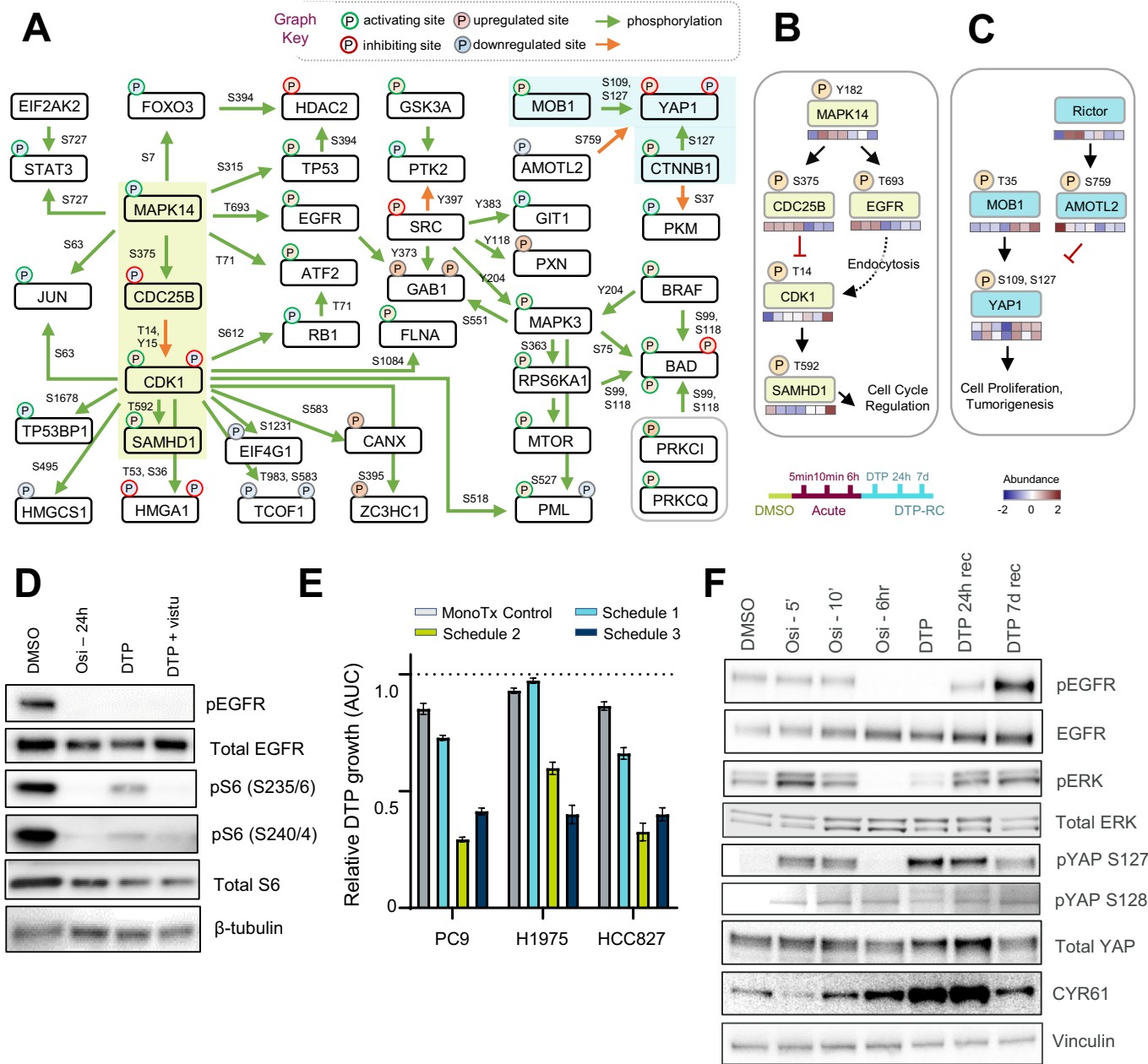

**Figure 7. Systematic phosphorylation model of signaling pathway regulations contributing to drug tolerance.**

(A) CausalPath results of differentially phosphorylation profiling between acute and DTP recover 24 h. (B, C) Signaling pathway regulation observed in the DTP recovery model is also reflected in the phosphoproteomic profiles. (D) PC9 cells were treated with osimertinib (160 nM) for 24 h, 14 days (DTP), or 11 days with osimertinib then 3 days with osimertinib + vistusertib (100 nM). (E) PC9, H1975, and HCC827 cells were treated long-term with the varying schedules of osimertinib (160 nM) and/or vistusertib (100 nM), as described in Fig. 6C. Growth was assessed as % confluence as measured by the Incucyte imaging platform. Bar graphs represent the normalized AUC of the plotted confluence. Error bars represent SD (n = 3). (F) PC9 cells were treated with osimertinib (160 nM) for the indicated time points, lysed and subjected to western blotting using the indicated antibodies. Source data are available online for this figure.

the human genome (93%) (Omenn et al, 2024), our DTP spectral library presents a valuable resource for lung cancer models and can be extended to other cancer and sample types, offering the potential to serve as a more versatile and widely applicable tool for DIA analysis. However, it is important to acknowledge the limitations of applying a single spectral library across diverse biological or technical contexts. The applicability of such libraries may be affected by sample-specific

proteome composition, as well as variations in fragmentation patterns due to differences in mass spectrometers or acquisition settings. As such, this resource should not be viewed as a one-size-fits-all solution, and careful consideration is needed to ensure appropriate use and to minimize false discoveries. By making these spectral library datasets freely available, we believe this mass spectra digital map combined with the DIA-MS strategy will provide a comprehensive resource for

advancing proteomics and phosphoproteomics applications in cancer research.

Our work reveals the intriguing discovery that CDK1 is potentially a critical factor in regulating the DTP phenotype. We identified several CDK1 substrates showing enhanced phosphorylation levels in the DTP state and validated the dynamic increase of SAMHD1 phosphorylation. SAMHD1 has been shown to play a critical role in maintaining dNTP pools to support DNA repair (Coggins et al, 2020), and its activation may allow cells to cope with the stress of prolonged drug treatment. There is an increasing understanding that DTPs are dependent on an intact DNA damage response (Ali et al, 2022). We did find that specific targeting of SAMHD1 by siRNA-mediated knockdown had no effect on DTP growth, indicating that it is the coordinated action of multiple CDK1 substrates that may be driving the DTP phenotype.

While it has been implicated in diverse cellular processes, the core function of CDK1 is to regulate the G2/M transition in the cell cycle (Massacci et al, 2023). Initial studies of DTPs worked under the assumption that this tolerant phenotype was a dormant or quiescent state of cell cycle arrest (Mikubo et al, 2021; Sharma et al, 2010). It has since been shown that a significant subset of DTPs do indeed progress through the cell cycle, albeit at a slower rate than untreated parental cells (Oren et al, 2021). A more recent study using a fluorescent tracker to quantify CDK2 activity demonstrated that while acute osimertinib treatment does drive a vast majority of cells into a quiescent, CDK2-low state, within days a subset of "escaper" cells with high CDK2 activity emerge (Hoffman et al, 2023). Given the overlap in cell cycle function, it would be interesting to test if CDK1 undergoes a similar temporal activation. In the clinic, recurring genetic amplifications of CDK4, CDK6, and multiple cyclin genes have been identified in osimertinib-resistant patients (Chmielecki et al, 2023), highlighting the importance of maintaining cell cycle progression to overcome drug treatment.

Another interesting finding is the dynamic regulation of the transcriptional coactivator YAP1 in the DTP model. YAP1 phosphorylation on Ser127 can lead to inhibitory 14-3-3 interactions and sequestration in the cytoplasm. Nevertheless, phosphorylation by nemo-like kinase (NLK) at YAP1-Ser128 disrupts its interactions with 14-3-3 protein, allowing active YAP1 to accumulate in the nucleus. Indeed, we and others have previously shown that YAP1 shows increased nuclear localization in DTPs (Criscione et al, 2022; Kurppa et al, 2020), and we also observe enhanced protein expression of CYR61, a canonical YAP1 target gene, in DTPs from multiple cell lines (Appendix Fig. S9). The phosphorylation status of YAP1 has a significant impact on its regulation on various signaling pathways, linking it to key cellular processes such as cell growth, proliferation, and survival, all of which are critical in DTP progression and recovery. This highlights the potential of targeting YAP1 phosphorylation for therapeutic intervention (Hergovich, 2017; Leblanc et al, 2021). Our phosphoproteomics results strongly suggest that the phosphorylation of these two crucial YAP1 sites is fine-tuned during the establishment of the DTP phenotype. Further elucidation of these phosphorylation dynamics not only advances our understanding of the function of YAP1 in DTPs but also emphasizes the importance of additional exploration to explore these regulatory mechanisms as potential therapeutic targets and biomarkers in DTP progression. Hence, the comprehensive understanding of phosphorylated mediators such as YAP1 and the CDK1–SAMHD1 axis reveals critical insights into the cellular signaling mechanisms that govern DTP recovery. With the extensive DIA library database and unique properties of DTP models, our work sets a foundation for phosphoproteomics-driven discovery of novel targets and biomarkers for modulating DTP. While our findings mark significant progress in this area, further efforts are required to translate these insights into clinical practice.

Overall, these findings present several promising new targets that act as vulnerabilities in DTPs. Notably, our validation experiments confirmed that the phosphoproteomic changes observed in DTPs were maintained across multiple EGFR-mutant cell lines. One major limitation of intervening with combinations after genetically-driven osimertinib resistance has been established is the high degree of heterogeneity in resistance mechanisms, often limiting the effectiveness of targeted therapies in a small proportion of patients (Chmielecki et al, 2023). While identifying drug-tolerant cells in patients remains a substantial hurdle, combination treatments aimed at minimal residual disease (MRD) phenotype, such as those identified in this study, could achieve broader efficacy by eliminating the residual tumor cells that ultimately drive drug resistance.

# Methods

**Reagents and tools table**

| Reagent/resource | Reference or source | Identifier or catalog number |
| --- | --- | --- |
| **Experimental models** | | |
| PC9 Cell Line | ECACC | 90071810 |
| NCI-H1975 Cell Line | ATCC | CRL-5908 |
| HCC4006 Cell Line | ATCC | CRL-2871 |
| **Recombinant DNA** | | |
| **Antibodies** | | |
| β-actin | Cell Signaling | RRID:AB_10950489 |
| CDK1 (phospho-Y15) | Cell Signaling | RRID:AB_560953 |
| CDK1 total | Cell Signaling | RRID:AB_2074795 |
| CDK2 total | Cell Signaling | RRID:AB_2276129 |
| CYR61 total | Cell Signaling | RRID:AB_2798492 |
| EGFR (phospho-Y1068) | Cell Signaling | RRID:AB_2096270 |
| EGFR total | Cell Signaling | RRID:AB_2895042 |
| ERK (phospho T202/Y204) | Cell Signaling | RRID:AB_331646 |
| ERK total | Cell Signaling | RRID:AB_330744 |
| GAPDH | Cell Signaling | RRID:AB_561053 |
| PML (phospho S518) | Abcam | RRID:AB_11140861 |
| PML total | Abcam | Ab240213 |
| S6 (phospho S235/6) | Cell Signaling | RRID:AB_916156 |
| S6 (phospho S240/4) | Cell Signaling | RRID:AB_10694233 |
| S6 total | Cell Signaling | RRID:AB_2238583 |
| SAMHD1 (phospho T592) | Cell Signaling | RRID:AB_2800147 |
| SAMHD1 total | Cell Signaling | 76700 |
| tubulin beta | Cell Signaling | RRID:AB_2210545 |
| Vinculin | Millipore | RRID:AB_477629 |

| Reagent/resource | Reference or source | Identifier or catalog number |
|---|---|---|
| YAP (phospho S127) | Cell Signaling | RRID:AB_2218911 |
| YAP (phospho S128 | Antibodies.com | A306241 |
| YAP total | Cell Signaling | RRID:AB_560953 |
| **Oligonucleotides and other sequence-based reagents** | | |
| sgRNA target sequences | Integrated DNA Technologies | See methods for sequences |
| **Chemicals, enzymes, and other reagents** | | |
| Sequencing grade modified trypsin | Promega | Catalog: V5111 |
| Formic acid, OPTIMA LC/ MS GRADE | Fisher | Catalog: LS118-4 |
| Acetonitrile LC-MS Grade | Merck | Catalog: 1.00030.4000 |
| Ni-NTA superflow agarose beads | Qiagen | Catalog: 30410 |
| Iron (III) chloride | Sigma | Catalog: 451649 |
| Pierce™ BCA Protein Assay Kit | ThermoFisher | Catalog: 23225 |
| RPMI 1640 Media w/ Glutamax | Gibco | Catalog: 61870036 |
| Fetal Bovine Serum | Sigma | Catalog: F7524 |
| RIPA buffer | ThermoFisher Scientific | Catalog: 89900 |
| PhosSTOP phosphatase inhibitor tablets | Roche | Catalog: 04 906 837 001 |
| cOmplete protease inhibitor tablets | Roche | Catalog: 04 693 116 001 |
| NuPage LDS Sample Buffer | ThermoFisher Scientific | Catalog: NP0008 |
| NuPage Reducing agent | ThermoFisher Scientific | Catalog: NP0009 |
| Novex Sharp pre-stained protein standard | ThermoFisher Scientific | Catalog: LC5800 |
| NuPage MOPS SDS running buffer (20x) | ThermoFisher Scientific | Catalog: NP0001 |
| SuperSignal™ West Dura Extended Duration Substrate | ThermoFisher Scientific | Catalog: 34075 |
| Lipofectamine RNAi MAX | ThermoFisher Scientific | Catalog: 13778150 |
| Lentiviral particels: pKLV2-EF1a-Cas9Bsd-W | Addgene | Catalog: 68343 |
| Blasticidin | Sigma | Catalog: 203350 |
| **Software** | | |
| Spectronaut | Biognosys, v.14 | https://biognosys.com/software/spectronaut/ |
| Perseus 1.6.15.0 | Tyanova et al, 2016 | http://www.perseus-framework.org |
| KSEA | https://doi.org/10.1093/bioinformatics/btx415 | https://casecpb.shinyapps.io/ksea/ |
| **Other** | | |

## Cell culture and preparation

The human lung adenocarcinoma cell lines PC9, HCC4006, NCI-H1975 were obtained from American Type Culture Collection (ATCC), and were grown in RPMI 1640 medium containing 10% FBS, 2 mM L-glutamine, and 1% penicillin-streptomycin at 37 °C in a humidified atmosphere of 5% $CO_2$/95% air. PC9 cells treated with 160 nM osimertinib for different time points, were washed with phosphate-buffered saline (PBS, 0.01 M sodium phosphate, 0.14 M NaCl pH 7.4), the supernatant removed, and stored at −80 °C before shipping to Academia Sinica.

## Generation of CDK1 and CDK2 knockout cell lines

Stable Cas9 expressing cell lines of PC9, NCI-H1975, and HCC4006 were generated as described previously (Pfeifer et al, 2024). Briefly, PC9, HCC4006 and NCI-H1975 parental cell lines were transduced with lentivirus particles of pKLV2-EF1a-Cas9Bsd-W (Addgene #68343), and selected using Blasticidin. CDK1 and CDK2 knockout cell lines were generated by reverse transfecting synthetic guide RNAs (sgRNAs) targeting CDK1 and CDK2 (2 sequences per gene, denoted as KO#1 and KO#2), using Lipofectamine RNAiMAX (for 48 h). The knockout pools were validated using western blotting and expanded for further use in the DTP growth assays.

The sgRNAs target sequences ordered from the predesigned library of Integrated DNA Technologies (IDT)) used in this study are listed below:

CDK1 KO#1: 5'-GGGTTCCTAGTACTGCAATT-3'
CDK1 KO#2: 5'-GAAGAATCCATGTACTGACC-3'
CDK2 KO#1: 5'-TCATGGGTGTAAGTACGAAC-3'
CDK2 KO#2: 5'-AAGATCGGAGAGGGCACGTA-3'.

## DTP growth assay

Cells were plated in 48-well dishes (20,000 cells/well) and allowed to attach overnight, and subsequently treated in triplicate (technical replicates) with the indicated drugs and immediately placed on the Incucyte S3 bioanalyzer. Imaging reads were taken every 24 h to calculate cell confluence as an indicator of cell growth. Throughout the experiment, wells were washed 1× with PBS and replenished with fresh media containing the appropriate treatments 2× per week. Three dosing schedules were employed as described in the results section: 1. Upfront combinations *vs.* control osimertinib monotherapy with 10 days of treatment followed by washout in drug-free media, 2. Generation of established DTP cells with 10 days of osimertinib monotherapy treatment followed by a switch to the second test compound, 3. Generation of established DTP cells with 10 days of osimertinib monotherapy treatment followed by the addition of the second test compound while maintaining osimertinib. After an additional 10 days, drugs were washed out by treating with drug-free media, and growth was compared to a 20-day osimertinib monotherapy control.

## Protein extraction and tryptic digestion

Once received, cells were reconstituted in lysis buffer cocktail (1% SL buffer, 10 mM TCEP, 40 mM CAA, protease inhibitor, and

phosphatase inhibitors in 100 mM Tris pH 8.5). The lysed cells were heated to 95 °C for 5 min, sonicated at 4 °C for 30 min, and then centrifuged at $16,000 \times g$ for 20 min at 4 °C. The supernatant was collected and StageTip protein digestion was performed. The StageTip was prepared by packing three layers of reverse-phase SDB-RPS Empore disks in 1 mL pipette tips. Lysed protein samples were loaded into StageTip and digested for 16 h with Lys-C 1:100 (w:w, Lys-C:protein) and trypsin 1:50 (w:w, trypsin:protein). The digested sample was acidified with 0.5% TFA and then five volume of ethyl acetate were added and vortexed ($500 \times g$, 2 min, RT). The resultant peptides were washed first with 1:1 ethyl acetate:0.2%TFA (v:v) and then by centrifuging with 0.2% TFA ($1000 \times g$, 1 min, RT). Finally, the peptides were eluted with 80% ACN and transferred into a new tube.

## Phosphopeptide enrichment by Fe-IMAC

The phosphopeptide enrichment was performed by home-made immobilized metal affinity chromatography (Fe-IMAC) StageTip. The IMAC tip was capped at one end with a 20-μm polypropylene frits disk (Agilent, Wilmington, DE, USA) enclosed in a tip-end fitting. The tip was packed with 40 μL of Ni-NTA superflow agarose beads (Qiagen) for each sample, washed with water three times, and the supernatant was removed. All purification steps for buffer exchange and sample loading were performed by centrifugation. The Ni2+ ions were removed with 50 mM EDTA in 1 M NaCl, followed by washing the beads three times with water. To generate the Fe-NTA beads, 200 μL of 100 mM FeCl$_3$ was incubated with the above washed beads for 30 min, the mixture was washed three times with water and the supernatant was removed prior to sample loading. Tryptic peptides (typically 200 μg) were reconstituted in loading buffer and loaded onto the IMAC tip for 30 min incubation. After successive washes by 100 μL washing buffer (loading buffer:ACN = 3:1) and 0.5% acetic acid, the bound phosphopeptides were eluted twice from the IMAC tip with 150 μL of 500 mM potassium phosphate buffer and desalted using reversed-phase StageTips, dried in vacuo, and reconstituted in 0.1% FA for LC-MS/MS analysis.

## LC-MS/MS analysis

Thermo Scientific™ UltiMate™ 3000 RSLCnano system (Thermo-Fisher Scientific) coupled to an Orbitrap Fusion Lumos mass spectrometer (ThermoFisher Scientific) was used in this study. Tryptic peptides were resuspended in 0.1% FA and spiked with iRT peptides according to the manufacturer's protocol. Peptides were separated using Thermo Scientific™ PepMap™ C18 50 cm × 75 μm ID column (ThermoFisher Scientific) with a 4–24% ACN gradient (proteome analysis) or a 6–22% (phosphoproteome analysis) ACN gradient in 0.1% FA over 120 min at a flow rate of 250 nL/min. The MS DIA datasets were acquired using the following parameters: scan range = 400–1250 m/z, MS resolution of 120,000 for a maximum ion injection time of 50 ms with an AGC target value of 4e5. The MS/MS spectra were fragmented by HCD using a normalized collision energy of 30%. The isolation window = 15 Da with 1 Da overlap over 400–1000 *m/z* precursor and scan range of 110–1600 *m/z*, resolution 15,000 with maximum injection time of 22 ms, AGC target = 5e4. All data were acquired in positive polarity and profile mode.

## Hybrid phosphoproteome spectral library construction

The original in-house spectral libraries were constructed by Spectronaut (Biognosys, v.14) in our previous report (Kitata et al, 2021). Briefly, samples from non-small cell lung cancer (NSCLC) cell lines and tissue were lysed, digested, and fractionated by HPLC. Phosphopeptides were further enriched by iron-based immobilized metal affinity chromatography (Fe-IMAC) and analyzed with data-dependent acquisition (DDA) and data-independent acquisition (DIA) modes. For the lung cancer proteome library, all proteome DDA raw files were searched by MaxQuant with an FDR of 1% at PSM. To generate the hybrid phosphopeptide reference library, combined database search results from DDA and DIA datasets were performed by Spectronaut Pulsar, to ensure identification confidence of 1% FDR cutoff at PSM, peptide, and protein for the estimation of phosphosite localization probability. To customize the specific spectral libraries for this project, 36 DIA datasets from proteome and phosphoproteome analysis were included in the original hybrid libraries to construct peptide and phosphopeptide reference libraries.

## DIA data processing for proteins and phosphosites identification

All DIA raw files were processed with Spectronaut (Biognosys, v.14) using a library-based approach (libDIA) against the above reference proteome and phosphoproteome libraries. Phospho (STY) was added as a variable modification to the default settings, including Carbamidomethyl (C) as a fixed modification and Oxidation(M), Acetyl (Protein N-term) as variable modifications. The FDR was set to 1% at the peptide precursor and protein level using scrambled decoy generation and dynamic size at 0.1 fraction of the library size. MS2-based quantification was used, enabling local cross-run normalization. The phosphosite localization tool recently integrated into Spectronaut was applied to filter for class 1 localization (probability ≥0.75). To demonstrate the accuracy of DIA-based identification, representative extracted ion chromatograms (XICs) of key phosphosites with their corresponding fragment ion traces are shown (Appendix Fig. S10A), highlighting the consistency of transitions used for quantitation. The reproducibility of DIA-based quantitation was further evaluated by calculating the coefficient of variation (CV) across replicates, with the distribution presented in Appendix Fig. S10B.

## Bioinformatics analysis

Most of the bioinformatics analysis were performed using Perseus software (1.6.15.0) (Tyanova et al, 2016). All the protein and phosphopeptide abundances were calculated as logarithmic (log2) ratios. The unique protein groups were consolidated in a single table containing all samples (columns=samples, rows=proteins). The row mean was subtracted from each batch's protein to obtain log2-scaled values. Statistical significance of changes in abundance among different timepoints was calculated by paired two-sample *t* tests ($P < 0.05$). The principal component analysis (PCA) was performed using log2-scaled values as features to show similarities between our different time-point treatment sample sets. Pathway enrichment analysis was processed by KEGG and ShinyGO. Phosphosite-specific quantification was performed by in-house customized R scripts.

## Kinase enrichment analysis

The comprehensive kinase–substrate relationships annotated in the PhosphoSitePlus database (Hornbeck et al, 2015) were used to perform the kinase enrichment analysis from the differentially changed phosphopeptide substrates. Unique phosphorylation motifs were identified as a 15-mer sequence (±7 amino acids surrounding the phosphorylation site). The top 50 potential individual sites exhibiting DTP recovery patterns were selected and submitted to Perseus to annotate their corresponding upstream kinases.

## CausalPath network analysis

To discover novel causal networks from the high-throughput phosphoproteomic data, we used the CausalPath platform (Luna et al, 2021). First, measurement values were processed into the specific format of CausalPath, which required to associate the normalized value with related gene symbols, and to specify the phosphorylation sites of phosphopeptides by in-house customized R scripts. A total of 95,663 phosphorylated sites from time points in Acute and in DTP phases were filtered by removing the duplicate values, valid values, and NA sites, as well as setting the false discovery rate (FDR) at 0.1 cutoff, resulting in about 1500 sites that were further processed. To perform the analysis, we installed CausalPath locally using the open-source Java code (https://github.com/PathwayAndDataAnalysis/causalpath). After uploading the tab-delimited format of the phospho-proteomic data, defined parameters were adjusted to display the causal interactions of our dataset and prior causal findings from pathway databases.

## Reagents and western blotting

All small-molecule inhibitors were synthesized according to published methods. Capivasertib was discovered by AstraZeneca after a collaboration with Astex Therapeutics (and its partnership with the Institute of Cancer Research and Cancer Research Technology Limited). For western blotting, cultured cells were washed once in cold PBS. Cells were scraped into 100 μL RIPA lysis buffer [25 mM Tris-HCl (pH 6.8), 3 mM EDTA, 5 mM EGTA, 0.27 M sucrose 0.5% Triton X-100, 50 mM NaF, 2 mM $Na_3VO_4$, 10 mM β-glycerophosphate, 5 mmol/L sodium pyrophosphate, and complete protease inhibitor tablets (Roche)] per 35-mm dish. Protein concentrations were determined by the bicinchoninic acid (BCA) protocol (Pierce) and western blots were performed by running samples of equal protein concentration on SDS-PAGE gels (NuPAGE Novex 4%–12% Bis-Tris Protein Gel, Thermo Scientific), transferring proteins to nitrocellulose membranes, incubating with primary antibodies overnight, followed by addition of horseradish peroxidase (HRP)-conjugated secondary antibodies (Cell Signaling Technology: goat anti-rabbit RRID: RRID:AB_2099233 and goat anti-mouse RRID: AB_330924) and detected with SuperSignal West Dura chemiluminescent substrate (Thermo Scientific). All antibodies used are listed in the Reagents and Tools Table.

## Data availability

The mass spectrometry proteomics data have been deposited to the ProteomeXchange Consortium via the PRIDE (Perez-Riverol et al,

2022) partner repository with the dataset identifier PXD058009. The datasets will be made publicly available upon publication of the manuscript.

The source data of this paper are collected in the following database record: biostudies:S-SCDT-10_1038-S44320-025-00141-1.

## Peer review information

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

## Acknowledgements

The study was funded by AstraZeneca. AstraZeneca reviewed the publication, without influencing the opinions of the authors, to ensure medical and scientific accuracy, and the protection of intellectual property. YJC acknowledges the Key and Novel Therapeutics Development Program for Major Diseases (AS-KPQ-111-KNT) at Academia Sinica in Taiwan. We thank the Medicinal Chemistry and Analytical Core Facilities (AS-NBRPCF-111-201) in the Biomedical Translation Research Center located at National Biotechnology Research Park for mass spectrometry analysis. We thank Huan-Chi Chiu for assistance with DIA precursor identification and quantification. Part of Fig. 1 was created with Biorender.com.

## Author contributions

**Hsiang-En Hsu**: Investigation; Writing—original draft; Writing—review and editing. **Matthew J Martin**: Validation; Investigation; Methodology; Writing—original draft; Writing—review and editing. **Shao-Hsing Weng**: Formal analysis; Methodology. **Reta Birhanu Kitata**: Software; Investigation; Methodology. **Srikar Nagelli**: Validation; Methodology; Writing—original draft; Writing—review and editing. **Chiung-Yun Chang**: Formal analysis; Visualization. **Sonja Hess**: Supervision; Visualization; Writing—original draft; Project administration; Writing—review and editing. **Yu-Ju Chen**: Conceptualization; Data curation; Supervision; Investigation; Writing—original draft; Project administration; Writing—review and editing.

Source data underlying figure panels in this paper may have individual authorship assigned. Where available, figure panel/source data authorship is

listed in the following database record: biostudies:S-SCDT-10_1038-S44320-025-00141-1.

## Disclosure and competing interests statement

The authors declare the following financial interests, which may be considered as potential competing interests: MM, SN, and SH are employees of AstraZeneca and may own shares and/or restricted stock of AstraZeneca. SN is an AstraZeneca Postdoctoral Fellow. Chiung-Yun Chang was an employee of AstraZeneca during the conduct of the study and is an employee of Syncell Inc. upon submission of this article; Syncell played no role and made no contribution, financial or otherwise, to the work in this manuscript.

