## [Peer Review File · Molecular Systems Biology]

Phosphoproteomics of Osimertinib-Tolerant Persister Cells Reveals Targetable Kinase-Substrate Signatures

Hsiang-En Hsu, Matthew Martin, Shao-Hsing Weng, Reta Kitata, Srikar Nagelli, Chiung-Yun Chang, Sonja Hess, and Yu-Ju Chen

Corresponding author(s): Yu-Ju Chen (yujuchen@as.sinica.edu.tw) , Sonja Hess (sonja.hess@astrazeneca.com)

Review Timeline:

Submission Date:	2nd Dec 24
Editorial Decision:	21st Dec 24
Revision Received:	26th May 25
Editorial Decision:	5th Aug 25
Revision Received:	17th Aug 25
Accepted:	19th Aug 25

Editor: Jingyi Hou

Transaction Report:

21st Dec 2024

Manuscript Number: MSB-2024-12776

Title: Phosphoproteomics of Osimertinib-Tolerant Persister Cells Reveals Targetable Kinase-Substrate Signatures

Author: Hsiang-En Hsu

Matthew Martin

Shao-Hsing Weng

Reta Kitata

Srikanth Nagelli

Chiung-Yun Chang

Sonja Hess

Yu-Ju Chen

Dear Prof Chen,

Thank you for submitting your work to Molecular Systems Biology. We have now heard back from the three reviewers who agreed to evaluate your manuscript. As you will see below, the reviewers acknowledge that the presented findings and datasets seem potentially interesting. They raise however a series of concerns, which we would ask you to address in a major revision.

The reviewers' recommendations are relatively clear, so there is no need to reiterate the points listed below. All the issues raised by the reviewers need to be satisfactorily addressed. As you may already know, our editorial policy allows in principle a single round of major revision, and it is therefore essential to provide responses to the reviewers' comments that are as complete as possible. Please feel free to contact me in case you would like to discuss in further detail any of the issues raised by the reviewers.

On a more editorial level, we would ask you to address the following issues:

- Please provide a .docx formatted version of the manuscript text (including legends for main figures, EV figures and tables). Please make sure that the changes are highlighted to be clearly visible.
- Please provide individual production quality figure files as .eps, .tif, .jpg (one file per figure).
- Please provide a .docx formatted letter INCLUDING the reviewers' reports and your detailed point-by-point responses to their comments. As part of the EMBO Press transparent editorial process, the point-by-point response is part of the Review Process File (RPF), which will be published alongside your paper.
- Please note that all corresponding authors are required to supply an ORCID ID for their name upon submission of a revised manuscript.
- We replaced Supplementary Information with Expanded View (EV) Figures and Tables that are collapsible/expandable online (see examples in <http://msb.embopress.org/content/11/6/812>). A maximum of 5 EV Figures can be typeset. EV Figures should be cited as 'Figure EV1, Figure EV2' etc... in the text and their respective legends should be included in the main text after the legends of regular figures.

Additional Tables/Datasets should be labeled and referred to as Table EV1, Dataset EV1, etc. Legends have to be provided in a separate tab in case of .xls files. Alternatively, the legend can be supplied as a separate text file (README) and zipped together with the Table/Dataset file.

For the figures and tables that you do NOT wish to display as Expanded View figures, they should be bundled together with their legends in a single PDF file called *Appendix*, which should start with a short Table of Content. Each legend should be below the corresponding Figure/Table in the Appendix. Appendix figures and tables should be referred to in the main text as: "Appendix Figure S1, Appendix Figure S2, Appendix Table S1" etc. See detailed instructions regarding expanded view here: <https://www.embopress.org/page/journal/17444292/authorguide#expandedview>.

- Before submitting your revision, primary datasets (and computer code, where appropriate) produced in this study need to be deposited in an appropriate public database (see <http://msb.embopress.org/authorguide> - dataavailability <https://www.embopress.org/page/journal/17444292/authorguide#dataavailability>). Please remember to provide a reviewer password if the datasets are not yet public. The accession numbers and database should be listed in a formal "Data Availability" section (placed after Materials & Method) that follows the model below (see also <https://www.embopress.org/page/journal/17444292/authorguide#dataavailability>). Please note that the Data Availability Section is restricted to new primary data that are part of this study.

Data availability

-At EMBO Press, we ask authors to provide source data for the main figures. Our source data coordinator will contact you to discuss which figure panels we would need source data for and will also provide you with helpful tips on how to upload and organize the files.

- Our journal encourages inclusion of *data citations in the reference list* to directly cite datasets that were re-used and obtained from public databases. Data citations in the article text are distinct from normal bibliographical citations and should directly link to the database records from which the data can be accessed. In the main text, data citations are formatted as follows: "Data ref: Smith et al, 2001". In the Reference list, data citations must be labeled with "[DATASET]". A data reference must provide the database name, accession number/identifiers and a resolvable link to the landing page from which the data can be accessed at the end of the reference. Further instructions are available at .

- We updated our journal's competing interests policy in January 2022 and request authors to consider both actual and perceived competing interests. Please review the policy <https://www.embopress.org/competing-interests> and update your competing interests if necessary. Please use the heading "Disclosure statement and competing interests".

- All Materials and Methods need to be described in the main text using our 'Structured Methods' format. According to this format, the Methods section includes a Reagents and Tools Table (listing key reagents, experimental models, software and relevant equipment and including their sources and relevant identifiers) followed by a Methods and Protocols section describing the methods, ideally using a step-by-step protocol format. The aim is to facilitate adoption of the methodologies across labs.

Please download and fill our Reagents and Tools Table template (.docx), which you can find in our author guidelines: <https://www.embopress.org/page/journal/17444292/authorguide#structuredmethods>.

An example of a Method paper with Structured Methods can be found here: <https://www.embopress.org/doi/10.15252/msb.20178071>.

- Regarding data quantification:

Please ensure to specify the name of the statistical test used to generate error bars and P values, the number (n) of independent experiments (please specify technical or biological replicates) underlying each data point and the test used to calculate p-values in each figure legend. Discussion of statistical methodology can be reported in the materials and methods section, but figure legends should contain a basic description of n, P and the test applied.

Graphs must include a description of the bars and the error bars (s.d., s.e.m.).

- Please provide a "standfirst text" summarizing the study in one or two sentences (approximately 250 characters, including space), three to four "bullet points" highlighting the main findings and a "synopsis image" (550px width and 400-600 px height, PNG format) to highlight the paper on our homepage.

Here are a couple of examples:

<https://www.embopress.org/doi/10.15252/msb.20199356>

<https://www.embopress.org/doi/10.15252/msb.20209475>

<https://www.embopress.org/doi/10.15252/msb.209495>

When you resubmit your manuscript, please download our CHECKLIST (<https://www.embopress.org/pb-assets/embosite/EMBO%20Press%20Author%20Checklist-1642513524327.xlsx>) and include the completed form in your submission.

Please note that the Author Checklist will be published alongside the paper as part of the transparent process (<https://www.embopress.org/page/journal/17444292/authorguide#transparentprocess>).

If you feel you can satisfactorily deal with these points and those listed by the referees, you may wish to submit a revised version of your manuscript. Please attach a covering letter giving details of the way in which you have handled each of the points raised by the referees. A revised manuscript will be once again subject to review and you probably understand that we can give you no guarantee at this stage that the eventual outcome will be favorable.

I look forward to receiving your revised manuscript soon.

Kind regards,

Jingyi

Jingyi Hou, PhD
Senior Editor
Molecular Systems Biology

We realize that it is difficult to revise to a specific deadline. In the interest of protecting the conceptual advance provided by the work, we recommend a revision within 3 months (21st Mar 2025). Please discuss the revision progress ahead of this time with the editor if you require more time to complete the revisions. Use the link below to submit your revision:

IMPORTANT: When you send your revision, we will require the following items:

1. the manuscript text in LaTeX, RTF or MS Word format
2. a letter with a detailed description of the changes made in response to the referees. Please specify clearly the exact places in the text (pages and paragraphs) where each change has been made in response to each specific comment given
3. three to four 'bullet points' highlighting the main findings of your study
4. a short 'blurb' text summarizing in two sentences the study (max. 250 characters)
5. a 'thumbnail image' (550px width and max 400px height, Illustrator, PowerPoint or jpeg format), which can be used as 'visual title' for the synopsis section of your paper.
6. Please include an author contributions statement after the Acknowledgements section (see <https://www.embopress.org/page/journal/17444292/authorguide>)
7. Please complete the CHECKLIST available at (<https://bit.ly/EMBOPressAuthorChecklist>). Please note that the Author Checklist will be published alongside the paper as part of the transparent process (<https://www.embopress.org/page/journal/17444292/authorguide#transparentprocess>).
8. When assembling figures, please refer to our figure preparation guideline in order to ensure proper formatting and readability in print as well as on screen:

See also figure legend guidelines: <https://www.embopress.org/page/journal/17444292/authorguide#figureformat>

9. Please note that corresponding authors are required to supply an ORCID ID for their name upon submission of a revised manuscript (EMBO Press signed a joint statement to encourage ORCID adoption). (<https://www.embopress.org/page/journal/17444292/authorguide#editorialprocess>)
Currently, our records indicate that the ORCID for your account is 0000-0002-3178-6697.

Link Not Available

11. Include a Reagents and Tools Table as part of the Methods section, which can be downloaded from our author guidelines (<https://www.embopress.org/page/journal/17444292/authorguide#structuredmethods>)

*** PLEASE NOTE *** As part of the EMBO Press transparent editorial process initiative (see our Editorial at <https://dx.doi.org/10.1038/msb.2010.72>), Molecular Systems Biology publishes online a Review Process File with each accepted manuscripts. This file will be published in conjunction with your paper and will include the anonymous referee reports, your point-by-point response and all pertinent correspondence relating to the manuscript. If you do NOT want this File to be published, please inform the editorial office at msb@embo.org within 14 days upon receipt of the present letter.

Reviewer #1:

Summary

In this manuscript, Hsu and colleagues use proteomic and phosphoproteomics approaches to investigate signaling in cells that

are deemed to be drug-tolerant persister (DTP) to Osimertinib, an EGFR inhibitor currently used to treat lung cancer. To this end, the authors first created spectral libraries consisting of several thousand peptides and phosphopeptides which were then used as the basis of DIA MS. These resources, it is claimed, will be a useful resource to the proteomics community that add value to the present study. This is in addition to the biological findings in the paper, which include the identification of proteins and signaling pathways, such as those driven by CDK1, MTOR and YAP1, that potentially contribute to drug resistance in DTP cells.

General remarks

Although the study and approach are of potential interest, there are inconsistencies in how the data are presented and interpreted, and questions about the quality and availability of the proposed resource. These issues should be addressed before considering the paper for additional review.

Major points

1. The authors use an approach described by Criscione et al., 2022 to generate DTP cells. However, this manuscript does not show that the response of cells to Osimertinib has changed after 21 days of prolonged exposure and the recovery time points (24h and 7 days). In other words, the authors should confirm that these cells are indeed DTP and have acquired tolerance to Osimertinib and how the recovery period affects this.
2. In section 3.5, it is shown that the presumably DTP cells increase phosphorylation of YAP1 at Ser127 and decreased phosphorylation in AMOTL2 at Ser759. In addition, the authors claim that phosphorylation of YAP1 at Ser127 downregulates AMOTL2 activity (Fig 7C). However, AMOTL2 is a regulator of YAP1 activity. The cited paper to support the idea that YAP1 modulates the activity of AMOTL2 actually shows that AMOTL2 regulates YAP1. Dephosphorylation of AMOTL2 at Ser759 (as described in the cited paper Artinian et al, 2015) inhibits YAP1 via a transcriptional mechanism. Furthermore, the phosphorylation of MOB1 at Thr35 enhances MOB1 binding to LATS1 leading to an increase in LATS1 kinase activity, which in turn phosphorylates and inhibits YAP1 activity (PMID: 17974916, PMID: 18328423). These results suggest an increase in Hippo pathway signaling leading to an inactivation of YAP1 by phosphorylation. The authors should revise the conclusions derived from this section.
3. Some of the phosphorylation events measured and shown in Fig 4 are inhibitory, but the authors use phosphorylation as if this had the same meaning as activation. For example, BRAF at S365 is inhibitory (PMID: 10869359). Similarly, the sentence [BAD] "exhibited constitutive activation through Serp99and Ser-p118 phosphorylation" is confusing because it is not clear that these phosphorylation sites activate BAD as most phosphorylation events on this protein are inhibitory (PMID: 9346240).
4. Related to the previous point, the authors claim that YAP1 is activated by treatment. However, the increased phosphorylation of YAP1 at Ser127 observed by the authors is usually linked to inactivation of YAP1 via 14-3-3 binding, cytoplasmic sequestration and degradation.
5. The proteomics and the phosphoproteomics analyses were performed in DTP PC9 cells and in DTP PC9 cells after recovery at 24 h and 7 days. However, a different cell line (NCI-H1975) was used to validate the results of the phosphoproteomic data and no explanation for using different models is provided in the paper. Have the authors checked whether CYR61 and other YAP1/TEADs linked genes are not changing in the proteomics data set for PC9 cells?
6. YAP1 binds and modulates the transcriptional activity of TEAD, which includes the expression of genes such as CYR61 and CTGF. However, YAP1 is not the only factor that can modulate the transcriptional activity of TEADs. For example, VGLL4, the glucocorticoid receptor (GR), TCF4, and AP-1 can modulate TEADs and therefore regulate CYR61 gene levels (PMID: 26832411, PMID: 31289134, PMID: 24525233, PMID: 28051067). In addition, recently Wu et al. have shown that Neurofibromin 2 (NF2) can modulate TEAD activity and CYR61 levels independently of YAP1 (PMID: 38522513). Thus, to substantiate their claims, the authors must show that in their model, YAP1 is not degraded and remain active in the nucleus, and the transcriptional activity of TEADs is modulated by YAP1 and not by other modulators.
7. Regarding the construction of spectral libraries, this was achieved by assembling DDA data at 1% FDR. While this error rate is accepted as adequate in discovery experiments, it is more debatable whether using this error rate to construct a resource is sufficiently stringent. Based on the number of entries in the libraries, around 3500 and 2900 peptides and phosphopeptides are likely to be incorrectly identified in the libraries reported in this study. The issue is that this error will propagate in future studies and will add to the error rate of matching MS data to libraries in subsequent DIA experiments.
8. The sentence "offering new avenues for therapeutic intervention to overcome EGFR-TKI resistance" is an exaggeration. Data in the paper suggest targeting CDK1, mTOR and YAP, which are among the most studied proteins in the literature and far from being "new avenues for therapeutic intervention".
9. Another exaggeration relates to the claim that "these results highlight the critical roles of mTOR and YAP signaling in sustaining DTP survival and regrowth". This is not shown in the paper. The data may indicate that these proteins may be active in cells treated with Osimertinib, but the authors have not shown that they have a role in the survival of the persister cells.
10. Regarding the kinase enrichment analysis in Figure 5, this seems to be an analysis centered on kinase substrates, rather than kinases themselves, but without assessing the statistical significance of association between kinases and their substrates.

The authors should carry out a kinase substrate enrichment analysis as described in other publications (PMID: 23532336).

11. The PRIDE submission cannot be reviewed because the submission has not been made public and reviewer passwords have not been provided. Similarly, the authors claim that the spectral libraries have been made publicly available. However, there are not details in the paper on how these can be reviewed.

Minor points

12. In section 3.4, the authors also show that CDK1 and CDK2 knockout leads to further reduction in cell confluency (Fig 6E). They do not specify whether this was calculated against the negative control. This should be properly indicated in the figure legend to avoid any misinterpretation of the results.

13. Fig. 4A does not show what is claimed in the text of page 7: "EGFR inhibition also led to increased phosphorylation in key oncogenic signaling pathways (Fig. 4A)".

14. In Figure 4B, PIK3C2B is mentioned under the PI3K/AKT heading but this protein is a class II PI3K without a role in the PI3K/AKT pathway.

15. The phosphorylation sites measured in Figures 7D and 7F should be named in the respective figure panel.

16. For all the WBs, the authors should indicate how many times the experiments were carried out and how the statistical significance of the data was evaluated.

Reviewer #2:

The manuscript by Hsu et al. „Phosphoproteomics of Osimertinib-Tolerant Persister Cells Reveals Targetable Kinase-Substrate Signatures" is a comprehensive study of the molecular mechanisms driving drug resistance in non-small cell lung cancer. The authors performed deep phosphoproteome and proteome profiling using library-based data-independent acquisition mass spectrometry, establishing the differences between the non-treated and multiple time intervals osimertinib-treated cells (which became persisters). They identified pathways contributing to resistance and possible treatment points and confirmed these findings in multiple cell lines. Their conclusions are well justified. This study is a great resource pointing to the new directions in overcoming the drug tolerance. The comments point out minor points that should nevertheless be addressed.

Detailed comments:

1. Have the authors performed the normalization of the phosphoproteome to total proteome? (to know if the change in the phosphosite abundance is due to increased phosphorylation/kinase activity or to the increase in total protein abundance)
2. Section 3.3 and Figure 3. What was the reason for the choice of the potential targets which were first downregulated in the acute phase and then upregulated in the recovery phase (and not the other way round, which would also be a meaningful change).
3. Please define well the "in vivo" used in Figure 6 (not used in the main manuscript which states (in my opinion, correctly "in cell lines". For many people with the exception of biochemists, "in vivo" means "in the organism".
4. Figure 7 has the spelling mistake in the word "tolerance"

Reviewer #3:

Hsu et al. explore the protein and phosphoproteome changes central to the drug-tolerant persister (DTP) cell phenotype in EGFR-mutated non-small cell lung cancer. They generated DTP cells by prolonged treatment of PC9-cells with Osimertinib and compared the proteome and phosphoproteome of untreated PC9 cells, the acute phase of treatment, and cells in the DTP phase, and DTP-recovery phase. They perform deep profiling using DIA mass spectrometry facilitated by spectral libraries generated from 36 DIA datasets. The results reveal an average number of 5.200 proteins and 21500 phosphopeptides per sample. The proteome data reveals clear regulated clusters, such as increased ribosome biogenesis, cytoskeleton regulation, and translation during recovery of EGFR. The phosphoproteomics showed intricate phosphorylation-driven signaling and pinpointed elevated CDK1 in DTP growth and critical roles of mTOR and YAP signaling. This study is valuable for understanding DTP cells and their signaling in NSCLC cancer and the broader understanding of inhibitor resistance. The study supports mass spectrometry data well with western blots, growth assays, and knockout cell lines and puts identified phosphosites and proteome changes in a biological context. The study needs clarification on a few points, which are outlined below. After that, I think the study is a good candidate for Molecular Systems Biology.

Major points:

Cells treated with Osimertinib for 21 days, followed by a 7-day recovery, are compared to control cells. Are any detected changes expected due to the cells being in culture for 30 days compared to the control cells that have not been in culture for that long?

Please share the data corresponding to (supplemental) figures, for example, figure 2C-D. Knowing what exact phosphosites or proteins were identified in the KEGG pathway analysis is essential for understanding the results.

Can the use of Osimertinib be verified using the acute phase phosphoproteome? Identifying a few known regulated phosphosites upon treatment is a valuable control.

How was the clustering in Figure 3a performed? The Kegg responses seem to follow a perfect trend, which is interesting. Was the complete KEGG pathway extracted, and then the heatmap was created?

There is clear re-activation of EGFR-controlled phosphorylation pathways during DTP and DTP recovery. Are there any signs this is due to EGFR re-activation, or can claims be made about what signaling actors are responsible for reactivating these pathways?

DIA is well suited for this study. A known caveat of DIA is that it can be overconfident in precursor identifications, such as identifying precursors based on only one or two detected transitions. It would be good to see a few traces of important identified phosphosites or peptides to confirm DIA data quality. Additionally, a CV plot for quantification would be valuable.

The authors promote the usefulness of the mass spectrometry libraries for others to use. I agree with this statement. It is important to note that DIA libraries are used to minimize false discoveries and should ideally include only precursors present in the sample. If this library is used in an experiment, there will be very likely more precursors than present in the sample, as different cell lines and tissues will have distinct (phospho)proteins. Also, different fragmentation energies lead to different relative abundance of fragment ions, meaning that different mass spectrometers or fragmentation settings will lead to different fragmentation patterns than the library. Therefore, such a library is not a one-size-fits-all solution but needs to be used with care.

Minor points:

Could the authors please comment on what they believe the biological difference is between the constitutively activated events and non-constitutive events during DTP recovery?

Please explain how the data in Figure 6E was normalized. Was this relative to control cells?

mTOR inhibition is shown to be important in EGFR-mutant cell lines. Is this effect specific to EGFR-mutant cell lines?

In the paper, the MRD phenotype is explained. Please clarify if you think you are studying the MRD-phenotype. And if so, how can you know you are not already in the post-MRD phenotypic phase?

Reviewer #1:

Summary

In this manuscript, Hsu and colleagues use proteomic and phosphoproteomics approaches to investigate signaling in cells that are deemed to be drug-tolerant persister (DTP) to Osimertinib, an EGFR inhibitor currently used to treat lung cancer. To this end, the authors first created spectral libraries consisting of several thousand peptides and phosphopeptides which were then used as the basis of DIA MS. These resources, it is claimed, will be a useful resource to the proteomics community that add value to the present study. This is in addition to the biological findings in the paper, which include the identification of proteins and signaling pathways, such as those driven by CDK1, MTOR and YAP1, that potentially contribute to drug resistance in DTP cells.

General remarks

Although the study and approach are of potential interest, there are inconsistencies in how the data are presented and interpreted, and questions about the quality and availability of the proposed resource. These issues should be addressed before considering the paper for additional review.

Major points

1. The authors use an approach described by Criscione et al., 2022 to generate DTP cells. However, this manuscript does not show that the response of cells to Osimertinib has changed after 21 days of prolonged exposure and the recovery time points (24h and 7 days). In other words, the authors should confirm that these cells are indeed DTP and have acquired tolerance to Osimertinib and how the recovery period affects this.

Reply:

PC9 cells have been used widely in the field as a model of drug tolerance vs. EGFR tyrosine kinase inhibitors (e.g. Hata et al. Nat. Med. 2016, Kurppa et al. Can. Cell 2020). Moreover, we have shown in a recent publication (Martin et al. Can. Res. Comm. 2022) that DTPs released from drug for 8-11 days re-establish EGFR phosphorylation, and most importantly, regain sensitivity to osimertinib treatment (as measured by induction of apoptosis; Figure 1 in that publication), indicating the reversible nature of the drug-tolerant phenotype. Similarly, we see a restoration of full EGFR activity in our DTP models for three cell lines used in this study after 7 days of drug withdrawal (**Fig. 6A and Appendix Figure S5A and S5B**).

2. In section 3.5, it is shown that the presumably DTP cells increase phosphorylation of YAP1 at Ser127 and decreased phosphorylation in AMOTL2 at Ser759. In addition, the authors claim that phosphorylation of YAP1 at Ser127 downregulates AMOTL2 activity (Fig 7C). However, AMOTL2 is a

regulator of YAP1 activity. The cited paper to support the idea that YAP1 modulates the activity of AMOTL2 actually shows that AMOTL2 regulates YAP1. Dephosphorylation of AMOTL2 at Ser759 (as described in the cited paper Artinian et al, 2015) inhibits YAP1 via a transcriptional mechanism. Furthermore, the phosphorylation of MOB1 at Thr35 enhances MOB1 binding to LATS1 leading to an increase in LATS1 kinase activity, which in turn phosphorylates and inhibits YAP1 activity (PMID: 17974916, PMID: 18328423). These results suggest an increase in Hippo pathway signaling leading to an inactivation of YAP1 by phosphorylation. The authors should revise the conclusions derived from this section.

Reply:

Thank you for your insightful comments to point out the inconsistency in our interpretation of AMOTL2 and YAP1 regulation. We acknowledge that AMOTL2 is a regulator of YAP1 activity, as described in Artinian et al. (2015), and we appreciate the clarification regarding the role of MOB1 in Hippo pathway activation. In response to your comment, we have revised Figure 7A (top right) to correctly depict the regulation of YAP1 by AMOTL2. Additionally, we have revised Figure 7C to incorporate Rictor as a cofactor of mTOR2 and to reflect the corrected regulatory pathway of AMOTL2 and YAP1. The manuscript is revised accordingly.

Figure 7

Revision: Page 8, Line 10-14: “...the result also showed an independent network of YAP1-127 phosphorylation of the Hippo pathway components, which may be either activated by phosphorylation of MOB1 at Thr35 (Zeng & Hong, 2008; Zhao et al, 2007) or downregulated by AMOTL2-pS759, causing the promotion of cell proliferation and tumorigenesis (Artinian et al, 2015) (Fig. 7C, also shown in blue box of Fig. 7A).”

3. Some of the phosphorylation events measured and shown in Fig 4 are inhibitory, but the authors use phosphorylation as if this had the same meaning as activation. For example, BRAF at S365 is inhibitory (PMID: 10869359). Similarly, the sentence [BAD] "exhibited constitutive activation through Serp99 and Ser-p118 phosphorylation" is confusing because it is not clear that these phosphorylation sites activate BAD as most phosphorylation events on this protein are inhibitory (PMID: 9346240).

Reply:

Thank you for pointing out the concern regarding the interpretation of phosphorylation events and their functional implications. After carefully reviewing the phosphorylation sites identified in the manuscript and relevant literature, we would like to provide the following clarifications:

1. **BRAF S365 Phosphorylation:** As the reviewer correctly pointed out, phosphorylation of BRAF at S365 has been shown to be inhibitory (PMID: 10869359). We have revised the manuscript to accurately reflect that phosphorylation at this site inhibits BRAF activity, and we have updated the legend of Figure 4 and text accordingly to clarify this negative regulation.

Revision: Page 6, Line 16-22: *"Among these complex phosphorylation events, some, such as GAB1-Y373, mTOR_S2481, and PKA_T198, are potentially associated with the activation of their respective pathways (PI3K/AKT, MAPK, and PKA). Thus, reduced phosphorylation of these phosphosites suggests their potential utility as biomarker to monitor the efficacy of osimertinib therapy. In contrast, phosphorylation of BRAF at S365 has been shown to inhibit its activity (Guan, 2000). These distinct phosphorylation events, leading to either activation or inhibition, highlight the dynamic regulation of NSCLC signaling in response to drug treatment."*

2. **BAD Serp99, Ser-p118 Phosphorylation:** We appreciate the literature that most phosphorylation events on BAD are inhibitory. However, we previously reported that GPER-mediated PKA activation leads to BAD Ser118 phosphorylation, which plays the key role in maintaining the stemness of breast cancer stem cells (PMID: 32421855). In this study, we observed that the temporal profile shows that Ser_p99 and Ser_p118 phosphorylation occur concomitantly with the upregulation of PKA and PKC during the DTP phase. However, their functional interplay potentially contributing to the observed anti-apoptotic phenotype is beyond the scope of current study. To improve the clarity, we have revised accordingly.

Revision: Page 6, Line 25-29: *"While BAD phosphorylation is generally associated with inhibition of its pro-apoptotic function (Datta et al, 1997), the increased phosphorylation of PKA and BAD align with our previous report that GPER-mediated PKA activation leads to BAD Ser118 phosphorylation. Thus, these events may contribute to the activation of anti-apoptotic processes, in line with the observed survival advantage of these cells (Qian et al, 2022)."*

3. Other Phosphorylation Sites:

GAB1-Y373 and **mTOR_S2481**: Based on literature and our analysis, phosphorylation at these sites is potentially associated with activation in the PI3K/AKT signaling pathway. **PKA_T198**: This phosphorylation site has been shown to be involved in the activation of PKA, which in turn regulates downstream signaling events. We have added this information to the revised manuscript to associate their potential role with the enriched signaling pathway.

Revision: Page 6, Line 16-20: *“Among these complex phosphorylation events, some, such as GAB1-Y373, mTOR_S2481, and PKA_T198, are potentially associated with the activation of their respective pathways (PI3K/AKT, MAPK, and PKA). Thus, reduced phosphorylation of these phosphosites suggest their potential utility as biomarker to monitor the efficacy of osimertinib therapy.”*

4. Related to the previous point, the authors claim that YAP1 is activated by treatment. However, the increased phosphorylation of YAP1 at Ser127 observed by the authors is usually linked to inactivation of YAP1 via 14-3-3 binding, cytoplasmic sequestration and degradation.

Reply:

Increased YAP1_S127 phosphorylation usually links to YAP1 inactivation. However, in our DISCUSSION, we hypothesize that YAP1_S127 and YAP1_S128 phosphorylation may interact, resulting in YAP1 activation. We have now added new evidence that YAP1_S128 increases in tandem with S127 in DTPs derived from PC9 (Fig. 7F) and HCC4006 (Appendix Figure S9B). It is also clear that the increase in phosphorylation of Ser127 does not block increases in YAP-pathway activation, as we had observed increases in the canonical YAP-driven protein CYR61 in H1975 cells (Appendix Figure S9A). We now support this observation even further by showing similar upregulation of CYR61 in DTPs derived from PC9 (Fig. 7F) and HCC4006 (Appendix Figure S9B) cell lines. Finally, we underscore the observation that Ser127 phosphorylation of YAP does not block its ability to drive downstream transcription by showing osimertinib-induced increases in this phospho-site in NF2-knockout PC9 cells, which have robust induction of CYR61 in the same time frame (Appendix Figure S9C).

Revision: Page 8, Line 35-41: *“Moreover, we also observe increased Ser127 phosphorylation in DTPs generated from NF2 knockout cells (Appendix Figure S9C), which have been shown to have enhanced activity in YAP1-driven transcription (Pfeifer et al.,2024), indicating that this phosphorylation event does not completely block transcriptional activity. Together, our findings indicate that mTOR and YAP1 signaling may be active in osimertinib-treated cells, but further investigation is needed to fully evaluate their role in the survival and regrowth of persister cells.”*

5. The proteomics and the phosphoproteomics analyses were performed in DTP PC9 cells and in

DTP PC9 cells after recovery at 24 h and 7 days. However, a different cell line (NCI-H1975) was used to validate the results of the phosphoproteomic data and no explanation for using different models is provided in the paper. Have the authors checked whether CYR61 and other YAP1/TEADs linked genes are not changing in the proteomics data set for PC9 cells?

Reply:

Thank you for this suggestion, to improve the robustness of our findings in the H1975 cell line. As described above we now present western blot data for CYR61 in PC9 and HCC4006 cells. We have also repeated the signaling experiments to test the effect of mTOR inhibitors in PC9 and HCC4006 cells (Figures 7D and Appendix Figure S8A).

Revision: Page 8, Line 17-20; Line 24-25: *“Our analysis showed that phosphorylation of S6, an mTOR downstream target, was significantly elevated in DTP cells compared to cells treated acutely with osimertinib in both PC9 and HCC4006 cells (Fig. 7D and Appendix Figure S8A). Moreover, continuous co-treatment with vistusertib significantly delayed the rapid-onset resistance phenotype of the HCC4006 cell line (Appendix Figure S8B).*

6. YAP1 binds and modulates the transcriptional activity of TEAD, which includes the expression of genes such as CYR61 and CTGF. However, YAP1 is not the only factor that can modulate the transcriptional activity of TEADs. For example, VGLL4, the glucocorticoid receptor (GR), TCF4, and AP-1 can modulate TEADs and therefore regulate CYR61 gene levels (PMID: 26832411, PMID: 31289134, PMID: 24525233, PMID: 28051067). In addition, recently Wu et al. have shown that Neurofibromin 2 (NF2) can modulate TEAD activity and CYR61 levels independently of YAP1 (PMID: 38522513). Thus, to substantiate their claims, the authors must show that in their model, YAP1 is not degraded and remain active in the nucleus, and the transcriptional activity of TEADs is modulated by YAP1 and not by other modulators.

Reply:

After checking the candidates provided by reviewer, we only see the protein expression of glucocorticoid receptor protein (NR3C1) is upregulated in DTP phase, based on our proteomics dataset. However, we also provide evidence for increased YAP1 protein expression by western blot in PC9 (Fig. 7F) and HCC4006 (Appendix Figure S9B), which may be sufficient to overcome any negative consequences of Ser127 phosphorylation described above.

7. Regarding the construction of spectral libraries, this was achieved by assembling DDA data at 1% FDR. While this error rate is accepted as adequate in discovery experiments, it is more debatable whether using this error rate to construct a resource is sufficiently stringent. Based on the number of entries in the libraries, around 3500 and 2900 peptides and phosphopeptides are likely to be incorrectly identified in the libraries reported in this study. The issue is that this error will propagate in future studies and will add to the error rate of matching MS data to libraries in subsequent DIA

experiments.

Reply:

Thank you for raising this important comment regarding the 1% FDR threshold for spectral library construction. With the great strength of DIA-based proteomic strategy, we have reported a few papers for DIA-based quantitative proteomics, including large-scale tissue profiling (Kitata et.al. Nat. Commun. 2022), microscale-to-nanoscale (5000-10 cells) proteomics (Siyal et. al. Anal. Chem, 2021), single cell proteomics (Gebreyesus et.al. Nat. Commun. 2022) and more recently the first single cell phosphoproteomics (Muneer et.al. Adv. Sci. 2025). Our approach follows established workflows with stringent identification, quantitation and mass spectra library construction. Moreover, during library construction, only identifications passing the 1% FDR threshold were retained, and any low-confidence identifications (i.e., those exceeding 1% FDR) were removed. Unlike other software that requires manual filtering of FDR thresholds after library construction, Spectronaut allows users to set the FDR threshold at the beginning of the process. This ensures that only high-confidence identifications are included, and low-confidence proteins, peptides, and PSMs are automatically excluded, when the library is generated. As a result, our hybrid DIA spectral libraries are rigorously curated, minimizing the risk of propagating incorrect identifications in subsequent analyses. Finally, for the phosphopeptides mentioned in the manuscript, we use additional criteria to determine Class-1 phosphopeptides, which confidently confirmed the phosphorylation sites by fragmentation ions pattern, as has been described in the manuscript, **Page 12, Line 39 - Page 13, Line 15**.

8. The sentence "offering new avenues for therapeutic intervention to overcome EGFR-TKI resistance" is an exaggeration. Data in the paper suggest targeting CDK1, mTOR and YAP, which are among the most studied proteins in the literature and far from being "new avenues for therapeutic intervention".

Reply:

Thank you for the comment. We have revised the manuscript to clarify that the identified druggable targets, such as CDK1, mTOR, and YAP, have been reported in the literature. We would, however, like to highlight that enhanced activation of CDK1 signaling in EGFR inhibitor-tolerant cells has not been reported to data (to our knowledge), and represents a key novel finding. Nevertheless, we have rephrased the text to highlight these targets as potential therapeutic options that could be further explored to overcome EGFR-TKI resistance. This revision more accurately reflects their established role in the field while acknowledging their relevance in the context of our DTP study.

Revision: Page 9, Line 13-14: *"..... expanding insight on the role of these targets in osimertinib resistance and DTP biology, which may inspire potential areas for further therapeutic exploration."*

9. Another exaggeration relates to the claim that "these results highlight the critical roles of mTOR and YAP signaling in sustaining DTP survival and regrowth." This is not shown in the paper. The data

may indicate that these proteins may be active in cells treated with Osimertinib, but the authors have not shown that they have a role in the survival of the persister cells.

Reply:

Thank you for your comment. We have revised the statement to better reflect the experimental evidence and to avoid overinterpretation.

Revision: Page 8, Line 39-41: *“Our findings indicate that mTOR and YAP1 signaling may be active in osimertinib-treated cells, but further investigation is needed to fully evaluate their role in the survival and regrowth of persister cells.”*

10. Regarding the kinase enrichment analysis in Figure 5, this seems to be an analysis centered on kinase substrates, rather than kinases themselves, but without assessing the statistical significance of association between kinases and their substrates. The authors should carry out a kinase substrate enrichment analysis as described in other publications (PMID: 23532336).

Reply:

We appreciate the reviewer’s suggestion to perform kinase substrate enrichment analysis in Figure 5. We have added a new analysis for kinase substrate enrichment analysis (KSEA) as suggested, following the approach described in PMID: 23532336. This analysis allowed us to directly assess the statistical significance of kinase activity changes across different DTP phases.

Our new results (Figure 5A) reveal a dynamic regulation of CDK1 activity throughout the treatment phases. Specifically, CDK1 was found to be downregulated during the acute phase but progressively upregulated in the DTP phase and DTP recovery phase. This finding corroborates our initial conclusion that substrate enrichment analysis (Figure 5B) highlights the elevated phosphorylation of multiple CDK1 substrates in the DTP phase. Moreover, our functional validation confirms that pharmacological or genetic inhibition of CDK1-mediated SAMHD1 activation significantly impairs DTP cell growth and survival. We have included the new KSEA results in the revised manuscript (Figure 5A) and have updated the text accordingly to reflect these findings.

Revision: Page 6, Lines 38-40: *“By the kinase–substrate enrichment analysis (KSEA), we showed that CDK1 activity changes during the treatment phases, with an increase in the phosphorylation of several CDK1 substrates during the DTP phase (Fig. 5A)”*

11. The PRIDE submission cannot be reviewed because the submission has not been made public and reviewer passwords have not been provided. Similarly, the authors claim that the spectral libraries have been made publicly available. However, there are not details in the paper on how these can be reviewed.

Reply:

We apologize for this oversight. The Project accession and Reviewer access are as follows:

URL: <https://www.ebi.ac.uk/pride/login>

Project accession: PXD058009

Reviewer access token: OkiVmYmKxutR

Minor points

12. In section 3.4, the authors also show that CDK1 and CDK2 knockout leads to further reduction in cell confluency (Fig 6E). They do not specify whether this was calculated against the negative control. This should be properly indicated in the figure legend to avoid any misinterpretation of the results.

Reply:

The growth curves used to plot Figure 6E have been normalized with curves of non targeted control (NTC). The figure legends of all the growth curves (Fig. 6F and Appendix Figure S7B, S7C and S7D) are updated appropriately in the revised version to avoid confusion.

13. Fig. 4A does not show what is claimed in the text of page 7: "EGFR inhibition also led to increased phosphorylation in key oncogenic signaling pathways (Fig. 4A)".

Reply:

We appreciate the reviewer's careful evaluation of Figure 4A and the related discussion in the manuscript. Figure 4A presents a summarized overview of phosphorylation changes in key oncogenic signaling pathways following EGFR inhibition. The details of findings are shown in **Figures 4B-4D**. During the acute phase, these figures show that the majority of phosphorylation events are reduced, indicating an overall suppression of these pathways by the osimertinib treatment. However, as the cells transition into the DTP and DTP recovery phases, phosphorylation of key signaling components, such as GAB1, mTOR and PKA, gradually increases, suggesting pathway reactivation. To provide a clearer view of these temporal dynamics, the detailed phosphorylation patterns of individual pathway components are provided in Figures 4B-4D.

Revision: Page 6, Line 5-16: *During the acute phase, phosphorylation levels of most key components were reduced, indicating pathway suppression. However, as cells transitioned into the DTP and DTP*

recovery phases, phosphorylation gradually increased (Fig. 4A), suggesting reactivation of these pathways. The inhibition of EGFR activity significantly suppressed downstream phosphorylation cascades such as PI3K/AKT (GAB1 at S266, S355, Y373, and T503; mTOR at S2478 and S2481) (Fig. 4B and Appendix Figure S3B), PKA (PRKACA at T198), PKC (PRKCD at T507 and S645; PRKCG at T655; PRKCI at S247, T412, and Y419; PRKCQ at Y545; PRKCZ at Y417) (Fig. 4C), and MAPK (BRAF at T373; ARAF at S157 and S257; RPS6KA3 at Y226, Y234, and S369) (Fig. 4D and Appendix Figure S3A). Notably, BRAF-S365 is an inhibitory phosphorylation site known to suppress MAPK pathway activation. Although most phosphorylation events lead to pathway activation, having both activating and inhibitory modifications shows that the regulation process is complicated.

14. In Figure 4B, PIK3C2B is mentioned under the PI3K/AKT heading but this protein is a class II PI3K without a role in the PI3K/AKT pathway.

Reply:

Thank you for the clarification. We have revised Figure 4B.

15. The phosphorylation sites measured in Figures 7D and 7F should be named in the respective figure panel.

Reply:

Thanks for this suggestion. The phosphorylation sites have been named in the revised version in Fig. 7D and F.

16. For all the WBs, the authors should indicate how many times the experiments were carried out and how the statistical significance of the data was evaluated.

Reply:

Each western blot experiment shown has been performed independently in 3 EGFRm cell lines, and we see good/very good correlation between them.

Reviewer #2:

The manuscript by Hsu et al. „Phosphoproteomics of Osimeritinib-Tolerant Persister Cells Reveals Targetable Kinase-Substrate Signatures" is a comprehensive study of the molecular mechanisms driving drug resistance in non-small cell lung cancer. The authors performed deep phosphoproteome and proteome profiling using library-based data-independent acquisition mass spectrometry, establishing the differences between the non-treated and multiple time intervals osimeritinib-treated cells (which became persisters). They identified pathways contributing to resistance and possible treatment points and confirmed these findings in multiple cell lines. Their conclusions are well justified. This study is a great resource pointing to the new directions in overcoming the drug tolerance. The comments point out minor points that should nevertheless be addressed.

Detailed comments:

1. Have the authors performed the normalization of the phosphoproteome to total proteome? (to know if the change in the phosphosite abundance is due to increased phosphorylation/kinase activity or to the increase in total protein abundance)

Reply:

Thanks very much for this comment. This is a good point to clarify whether the regulation is via protein expression or site-specific phosphorylation. We have been working on phosphoproteomics methodology and application for years. In our experience, the observed phosphopeptide level is a better way to reflect overall effect on both protein expression and site-specific phosphorylation. Thus, we explore the alteration in both protein expression and site-specific phosphorylation and do not perform additional normalization. In most cases, the different phosphorylation site has different alteration. In the example of BAD, it has un-changed, down- and up-regulated sites. Thus, this manuscript reported both datasets without performing normalization of the phosphorylation by the protein expression.

2. Section 3.3 and Figure 3. What was the reason for the choice of the potential targets which were first downregulated in the acute phase and then upregulated in the recovery phase (and not the other way round, which would also be a meaningful change).

Reply:

We wish to study DTP cell signaling pathways. When we treat cells with osimertinib in the acute phase, osimertinib-responsive targets are downregulated. In our hypothesis, the targets that enable tolerance may have resumed protein expression or have overly activated phosphorylation in the

DTP recovery phase. Therefore, we chose these targets that were downregulated in the acute period and increased in DTP recovery phase.

3. Please define well the "in vivo" used in Figure 6 (not used in the main manuscript which states (in my opinion, correctly "in cell lines". For many people with the exception of biochemists, "in vivo" means "in the organism".

Reply:

We thank the reviewer for this astute observation and have corrected this to "*in vitro*".

4. Figure 7 has the spelling mistake in the word "tolerance"

Reply:

Thank you for pointing out this mistake. We have corrected the spelling of the word 'tolerance'.

Reviewer #3:

Hsu et al. explore the protein and phosphoproteome changes central to the drug-tolerant persister (DTP) cell phenotype in EGFR-mutated non-small cell lung cancer. They generated DTP cells by prolonged treatment of PC9-cells with Osimertinib and compared the proteome and phosphoproteome of untreated PC9 cells, the acute phase of treatment, and cells in the DTP phase, and DTP-recovery phase. They perform deep profiling using DIA mass spectrometry facilitated by spectral libraries generated from 36 DIA datasets. The results reveal an average number of 5,200 proteins and 21,500 phosphopeptides per sample. The proteome data reveals clear regulated clusters, such as increased ribosome biogenesis, cytoskeleton regulation, and translation during recovery of EGFR. The phosphoproteomics showed intricately phosphorylation-driven signaling and pinpointed elevated CDK1 in DTP growth and critical roles of mTOR and YAP signaling. This study is valuable for understanding DTP cells and their signaling in NSCLC cancer and the broader understanding of inhibitor resistance. The study supports mass spectrometry data well with western blots, growth assays, and knockout cell lines and puts identified phosphosites and proteome changes in a biological context. The study needs clarification on a few points, which are outlined below. After that, I think the study is a good candidate for Molecular Systems Biology.

Major points:

1. Cells treated with Osimertinib for 21 days, followed by a 7-day recovery, are compared to control cells. Are any detected changes expected due to the cells being in culture for 30 days compared to the control cells that have not been in culture for that long?

Reply:

Drug tolerant persister (DTP) is a state where a subpopulation of cells survive exposure to the drug and its selective pressure. Since the untreated control cells were not subjected to any drug induced selective pressure, we do not anticipate major alterations in signaling pathways within the untreated group solely due to prolonged culture duration.

The PC9 cell line is very robust and widely used in the EGFR-TKI research. The DTP model utilized in this work is very established (published firstly in Sharma et al. Cell 2010). There are many similar studies where DTP states were compared to control cell lines (RNA Seq) such as Kurppa et al. Cancer Cell 2020, Criscione et al. 2022.

2. Please share the data corresponding to (supplemental) figures, for example, figure 2C-D. Knowing what exact phosphosites or proteins were identified in the KEGG pathway analysis is essential for understanding the results.

Reply:

We have provided the detailed list in Fig. 2C&D in the Supporting information (Table EV1).

3. Can the use of Osimertinib be verified using the acute phase phosphoproteome? Identifying a few known regulated phosphosites upon treatment is a valuable control.

Reply:

Thanks for the important comment. In the first part of proteomics results, we use the proteomics data to identify the targets in response to the action of Osimertinib treatment. Following your suggestion, we have added the information in the revised manuscript. Osimertinib primarily inhibits mutant EGFR expression by specifically targeting the EGFR-T790M mutation site. Investigating known regulatory phosphosites through acute-phase phosphoproteomics can provide valuable guides to monitor the treatment efficacy. As shown in Fig. 4, we observed significant down-regulation of many phosphosites under the PI3K/AKT, PKC/PKA, and MAPK signaling. These phosphorylation regulatory sites may serve as biomarkers for osimertinib therapy

Revision: Page 6, Line 8-20. *“The inhibition of EGFR activity significantly suppressed downstream phosphorylation cascades such as PI3K/AKT (GAB1 at S266, S355, Y373, and T503; mTOR at S2478 and S2481) (Fig. 4B and Appendix Figure S3B), PKA (PRKACA at T198), PKC (PRKCD at T507 and S645; PRKCG at T655; PRKCI at S247, T412, and Y419; PRKCQ at Y545; PRKCZ at Y417) (Fig. 4C), and MAPK (BRAF at T373; ARAF at S157 and S257; RPS6KA3 at Y226, Y234, and S369) (Fig. 4D and Appendix Figure S3A). Notably, BRAF-S365 is an inhibitory phosphorylation site known to suppress MAPK pathway activation. Although most phosphorylation events lead to pathway activation, having both activating and inhibitory modifications shows that the regulation process is complicated. Among these complex phosphorylation events, some, such as GAB1-Y373, mTOR_S2481, and PKA_T198, are potentially associated with the activation of their respective pathways (PI3K/AKT, MAPK, and PKA). Thus, reduced phosphorylation of these phosphosites suggests their potential utility as biomarker to monitor the efficacy of osimertinib therapy.”*

4. How was the clustering in Figure 3a performed? The Kegg responses seem to follow a perfect trend, which is interesting. Was the complete KEGG pathway extracted, and then the heatmap was created?

Reply:

As described in the manuscript, we first calculated the protein expression levels across the three acute-phase samples to obtain a baseline acute-phase protein expression level. Subsequently, we normalized the expression levels of each protein in the three DTP-phase conditions (DTP, DTP-24h, and DTP-7d) against this acute-phase average, followed by quantitative analysis to filter the differentially expressed proteins (DEPs) in the three DTP phases. Next, supervised clustering was performed for these DEPs using Perseus software, classifying the proteome into six distinct clusters. Proteins within each cluster were then subjected to KEGG pathway analysis to find their biological

pathways.

5. There is clear re-activation of EGFR-controlled phosphorylation pathways during DTP and DTP recovery. Are there any signs this is due to EGFR re-activation, or can claims be made about what signaling actors are responsible for reactivating these pathways?

Reply:

As shown in Figure 6A, during the DTP recovery phase, there is a significant increase in EGFR phosphorylation, indicating its reactivation. Additionally, we also identified an interesting regulatory pathway: MAPK14-CDC25B-CDK1-SAMHD1. This pathway was further validated through CDK inhibition experiments (Figure 6E-F), underscoring the critical role of CDK1-mediated downstream signaling in driving pathway activation. These findings highlight the important role of CDK1 in shaping signaling dynamics during the DTP model.

6. DIA is well suited for this study. A known caveat of DIA is that it can be overconfident in precursor identifications, such as identifying precursors based on only one or two detected transitions. It would be good to see a few traces of important identified phosphosites or peptides to confirm DIA data quality. Additionally, a CV plot for quantification would be valuable.

Reply:

Thank you for the insightful feedback. We have provided the following data in the revised manuscript.

1. Select example of precursors
2. CV distribution in quantitative performance

We would like to clarify that at least 3-5 fragments are used for DIA-based quantitation analysis. To address the concern, we have provided example XIC of key identified phosphosites or peptides to confirm the quality of the DIA data (Appendix Figure S10A). These traces highlight the transitions used for identification and their consistency. Additionally, we have added a CV plot for quantification to demonstrate the reproducibility and reliability of the data (Appendix Figure S10B). This should provide further validation of the DIA-based results and address concerns regarding overconfidence in precursor identifications.

7. The authors promote the usefulness of the mass spectrometry libraries for others to use. I agree with this statement. It is important to note that DIA libraries are used to minimize false discoveries and should ideally include only precursors present in the sample. If this library is used in an experiment, there will be very likely more precursors than present in the sample, as different cell lines and tissues will have distinct (phospho)proteins. Also, different fragmentation energies lead to different relative abundance of fragment ions, meaning that different mass spectrometers or fragmentation settings will lead to different fragmentation patterns than the library. Therefore, such

a library is not a one-size-fits-all solution but needs to be used with care.

Reply:

Thank you very much for this insightful comment. We fully agree with your assertion that the spectra library-based DIA is critically affected by a few factors: type of instrument (different ionization to generate different precursor, different fragmentation pattern, sample types, and computation approach with different FDR control. We have added these potential constraints in the Discussion.

Revision: Page 9, Lines 26-32: *“it is important to acknowledge the limitations of applying a single spectral library across diverse biological or technical contexts. The applicability of such libraries may be affected by sample-specific proteome composition, as well as variations in fragmentation patterns due to differences in mass spectrometers or acquisition settings. As such, this resource should not be viewed as a one-size-fits-all solution, and careful consideration is needed to ensure appropriate use and to minimize false discoveries.”*

Minor points:

1. Could the authors please comment on what they believe the biological difference is between the constitutively activated events and non-constitutive events during DTP recovery?

Reply:

We believe that the biological differences between constitutively activated events and non-constitutive events during DTP recovery lie in their distinct roles in cellular adaptation. Constitutively activated events likely represent core survival mechanisms, such as DNA repair, RNA splicing, and anti-apoptotic pathways, including YAP1 and mTOR-BAD hyperphosphorylation, which are essential for maintaining cellular viability during recovery. In contrast, non-constitutive events may reflect transient adaptive processes, such as increased ribosome synthesis and protein translation observed during the DTP phase. These events may serve as intermediate steps to support recovery until more robust resistance mechanisms are established.

2. Please explain how the data in Figure 6E was normalized. Was this relative to control cells?

Reply:

The growth curves used to plot Figure 6E have been normalized to the non-targeted control (NTC). The figure legends of all the growth curves (Fig. 6F and Appendix Figure S7B, S7C and S7D) are updated appropriately in the revised version to avoid confusion.

3. mTOR inhibition is shown to be important in EGFR-mutant cell lines. Is this effect specific to EGFR-mutant cell lines?

Reply:

Our phosphoproteomics data reveal that mTOR pS2481, an activator of phospho S6, is upregulated. Figure 7D and Appendix Figure S8A show that pS6 activity is restored in DTP samples compared to those acutely treated with osimertinib, suggesting mTOR pathway activation as a resistance mechanism. Treatment with vitsusertib, an mTOR1/2 inhibitor, abrogates the restored pS6 activity and DTP regrowth in combination assays.

These results are consistent across two EGFR mutant NSCLC cell lines, PC9 and HCC4006, highlighting their significance in EGFR mutant contexts, although experiments in non-EGFR mutant settings were beyond the scope of this study.

4. In the paper, the MRD phenotype is explained. Please clarify if you think you are studying the MRD-phenotype. And if so, how can you know you are not already in the post-MRD phenotypic phase?

Reply:

The DTP phenotype has been established to model the minimal residual disease (MRD) phenotype in the clinic. MRD is a state in which some cells survive the disease and cause relapse on drug holiday. Similarly, DTP cells on drug washout return to a proliferative state and regain drug sensitivity (Ramirez et al. Nat. Commun. 2016; Pu et al. Nat Rev Clin Oncol 2023)

We have shown in a recent publication (Martin et al. Can. Res. Comm. 2022) that DTPs released from drug for 8-11 days re-establish EGFR phosphorylation, and most importantly, regain sensitivity to osimertinib treatment (as measured by induction of apoptosis; Figure 1 in that publication), indicating the reversible nature of the drug-tolerant phenotype. Similarly, we see a restoration of full EGFR activity in our DTP models for three cell lines used in this study after 7 days of drug withdrawal (Fig. 6A and Appendix Figure S5A and S5B). Based on these, we are confident that the phenotype we see in our cells are indicative of DTP phenotype which mimics MRD in the clinic.

5th Aug 2025

Manuscript Number: MSB-2024-12776R

Title: Phosphoproteomics of Osimertinib-Tolerant Persister Cells Reveals Targetable Kinase-Substrate Signatures

Author: Hsiang-En Hsu

Matthew Martin

Shao-Hsing Weng

Reta Kitata

Srikar Nagelli

Chiung-Yun Chang

Sonja Hess

Yu-Ju Chen

Dear Prof Chen,

Thank you for sending us your revised manuscript. We have now heard back from the three reviewers who agreed to evaluate your revised study. As you will see below, the reviewers are satisfied with the performed revisions and support publication. Before we can formally accept the manuscript for publication, we would ask you to address some remaining editorial-level issues listed below.

1. Please provide each figure as a separate file in .eps, .tif, or .jpg format. Figures should not be submitted in PowerPoint (.pptx) format.

2. Appendix:

- Please add callouts for Appendix Figure S10.

- The title page should contain "Appendix for [manuscript title]" and a Table of Contents listing all included items along with their corresponding page numbers.

3. Appendix Table S1 needs to be updated and relabeled as Dataset EV1. Update all source file names, titles, legends, and manuscript callouts to reflect the new label (Dataset EV1). The dataset legend should be included as a separate sheet/tab within the Excel file.

4. Please rename "Conflict of interest" to "DISCLOSURE AND COMPETING INTERESTS STATEMENT".

5. Data availability: please provide specific URL for the PXD058009 dataset and remove the reviewer access token. Please ensure the datasets will be made publicly available upon the publication of the manuscript.

6. Remove "data not shown". As per our guidelines, on "Unpublished Data" the journal does not permit citation of "Data not shown". All data referred to in the paper should be displayed in the main or Expanded View figures.

7. Please address the following issues related to figure legends:

- Please note that the legend for figure 2D is missing in the manuscript. This needs to be rectified.

- Please indicate the statistical test used for data analysis in the legends of figures 3C, D

- Please note that information related to n is missing in the legends of figures 3C, D

- Please note that the error bars are not defined in the legends of figures 6D, F; 7E

8. Please revise the manuscript to follow the correct section order: Title page - Abstract & Keywords - Introduction - Results - Discussion - Methods - Data Availability - Acknowledgements - Disclosure and Competing Interests Statement - References - Figure Legends - Table(s) Expanded View Figure Legends.

When you resubmit your manuscript, please download our CHECKLIST (<https://bit.ly/EMBOPressAuthorChecklist>) and include the completed form in your submission. *Please note* that the Author Checklist will be published alongside the paper as part of the transparent process (<https://www.embopress.org/page/journal/17444292/authorguide#transparentprocess>)

Click on the link below to submit your revised paper.

Kind regards,
Jingyi

Jingyi Hou, PhD
Senior Editor
Molecular Systems Biology

*** PLEASE NOTE *** As part of the EMBO Press transparent editorial process initiative (see our Editorial at <https://dx.doi.org/10.1038/msb.2010.72> , Molecular Systems Biology will publish online a Review Process File to accompany accepted manuscripts. When preparing your letter of response, please be aware that in the event of acceptance, your cover letter/point-by-point document will be included as part of this File, which will be available to the scientific community. More information about this initiative is available in our Instructions to Authors. If you have any questions about this initiative, please contact the editorial office (msb@embo.org).

Reviewer #1:

The authors have addressed the comments made to the original submission.

Reviewer #2:

The authors have adequately addressed my comments.

Reviewer #3:

Thank you to the authors for their efforts to address my comments and concerns. They have discussed my concerns sufficiently through additional analyses, clear explanations in the manuscript, and through their responses. I believe the manuscript now meets the high standards of Molecular Systems Biology and is thus ready for publication.

All editorial and formatting issues were resolved by the authors.

19th Aug 2025

Manuscript number: MSB-2024-12776RR

Title: Phosphoproteomics of Osimertinib-Tolerant Persister Cells Reveals Targetable Kinase-Substrate Signatures

Dear Prof Chen,

Thank you again for sending us your revised manuscript. We are now satisfied with the modifications made and I am pleased to inform you that your paper has been accepted for publication.

Sincerely,
Jingyi

Jingyi Hou, PhD
Senior Editor
Molecular Systems Biology
